# Multi-Modal Language Models as Text-to-Image Model Evaluators

## Abstract

The steady improvements of text-to-image (T2I) generative models lead to slow deprecation of automatic evaluation benchmarks that rely on static datasets, motivating researchers to seek alternative ways to evaluate the T2I progress. In this paper, we explore the potential of multi-modal large language models (MLLMs) as evaluator agents that interact with a T2I model, with the objective of assessing prompt-generation consistency. We present **Multimodal Text-to-Image Eval (MT2IE)**, an evaluation framework that iteratively generates prompts for evaluation, scores generated images and matches T2I evaluation of existing benchmarks with a fraction of the prompts used in existing static benchmarks. We show that MT2IE's prompt-generation consistency scores have higher correlation with human judgment than prompt consistency metrics previously introduced in the literature. MT2IE generates prompts that are efficient at probing T2I model performance, producing the same relative T2I model rankings as existing benchmarks while evaluating on $80\times$ less prompts. We hope that these results will unlock the development of dynamic and interactive evaluation frameworks, and mitigate the deprecation of automatic evaluation benchmarks.

## 1 Introduction

With the ever-growing use of generated images in academic and commercial settings (Rombach et al., 2022; Podell et al., 2023; Sinha et al., 2024), the evaluation of text-to-image (T2I) generative models is a rapidly growing and highly relevant area of research. Aesthetic quality, diversity of generated samples, and image-text consistency are crucial for evaluating T2I models and driving the development of reliable, high-quality image generators. Image-text consistency seeks to measure how well a generated image matches its corresponding text prompt; its assessment has gained increasing attention as users' text prompts become longer and more semantically complex.

Generally, *static sets* of text prompts are used to benchmark T2I model progress. These prompts can be human-written, typically sourced from image captioning datasets (*e.g.*, COCO (Lin et al., 2014), Conceptual Captions (Changpinyo et al., 2021)), or automatically generated from free-form text, and designed to test image generation capabilities (*e.g.*, ImagenHub (Ku et al., 2023), ConceptMix (Wu et al., 2024), T2I-Compbench (Huang et al., 2023), HPDv2 (Wu et al., 2023b)). Evaluation frameworks that use static prompt datasets most commonly use automatic metrics to assess generated images. For image-text consistency evaluation, *alignment-based* or *generated question-answering-based* (GQA) metrics are commonly used. *Alignment-based* metrics judge the text consistency of the generated image by computing the similarity between the image and prompt. For example, CLIPScore (Hessel et al., 2021) leverages the CLIP model (Radford et al., 2021) to calculate cosine similarity between generated images and text prompts. However, its effectiveness is limited by the weaknesses of CLIP, which struggles with capturing compositional vision-language aspects such as spatial relationships, object cardinality, and attribute binding (Tong et al., 2024; Thrush et al., 2022b; Yuksekgonul et al., 2023). *GQA* metrics (Hu et al., 2023; Cho et al., 2024; Lin et al., 2024) have been introduced to address the limitations of fine-grained image-text consistency evaluation. These use large language models (LLMs) to generate question sets about the text prompt and then use visual question answering (VQA) models to score the generated questions given the prompt's generated image.

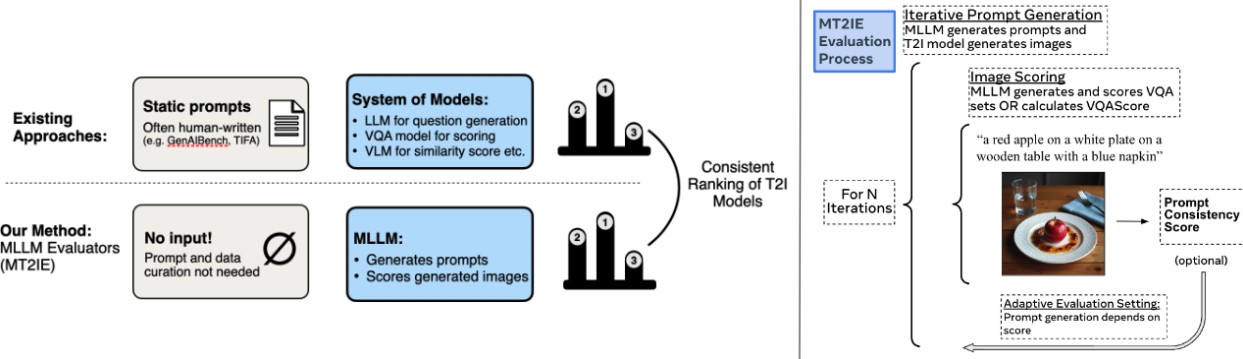

Figure 1: We present Multimodal Text-to-Image Eval (MT2IE), our novel evaluation methodology for text-to-image (T2I) generation. Previous approaches to T2I model evaluation require large, static benchmarks and often multiple external models. Our approach in a single system that both generates and scores prompts. We show that MT2IE can generate as few as 20 prompts while still maintaining consistent model rankings of other approaches that use more compute and more data.

Recently, multimodal LLMs (MLLMs) (Alayrac et al., 2022; Liu et al., 2023) have been developed to enhance large language models (LLMs) with vision understanding capabilities, resulting in state-of-the-art performance across some vision-language tasks. Various approaches use MLLMs for prompt consistency evaluation, often via some form of visual question answering (Lu et al., 2023; Ku et al., 2024; Zhang et al., 2023; Meng et al., 2024; Tan et al., 2024). Although these approaches leverage MLLMs for improved evaluation, they still require fixed prompt datasets without any ability for dynamic evaluation. This can create a narrow focus on optimizing T2I performance for predefined prompts, rather than driving broad improvements. Additionally, rapid advancements in model capabilities can render static prompt sets ineffective, as they become overly simplistic and fail to provide meaningful evaluation (Corry et al., 2021; Ott et al., 2022; Gupta et al., 2024). Across benchmarks and metrics, the use of *static evaluation frameworks* is prone to overfitting and vulnerable to deprecation.

In this work, we present **Multimodal Text-to-Image Eval (MT2IE)**, a unified framework that moves beyond static evaluation by iteratively generating dynamic prompts and scoring image generations. We move beyond the use of fixed prompt sets by exploring the potential of MLLMs as *evaluator agents* that interact with a T2I model. Instead of relying on a static benchmark of prompts for image generation, the MLLM is now responsible for both *generating* text prompts and *scoring* the generated images. This removes the need to curate a prompt set for benchmarking while performing end-to-end evaluation with a single model. This additionally allows us to leverage the MLLM to generate *increasingly complex* prompts for generation, pushing the limits of the T2I model, and *adaptive* prompts, automatically rewriting prompts according to the T2I model's performance.

In our approach, we focus on validating the capabilities of off-the-shelf MLLMs to evaluate image-text consistency. We start by assessing the potential of MLLMs to generate questions and score prompt-generation consistency of T2I models, showing that a single model can outperform existing human judgment correlation metrics on the static TIFA160 benchmark (Hu et al., 2023). This contrasts with previous work that either relied on proprietary GPT-4 variants (Zhang et al., 2023) or found open-source MLLMs' judgments ineffective and uncorrelated with human judgments (Ku et al., 2024; Jiang et al., 2024b). Then, we leverage MLLMs to not only score but also generate prompts of increasing complexity to evaluate T2I models. We use this MLLM-based approach to compare the prompt-generation consistency of 8 state-of-the-art T2I models and find that the rankings from our approach are comparable to those reported by static benchmarks, while only using a fraction (~1%) of prompts. Finally, we investigate MLLMs as evaluator *agents* for prompt-generation consistency by enabling interaction between the MLLM and the T2I model. Our contributions are as follows:

1. We present **Multimodal Text-to-Image Eval (MT2IE)**, an evaluation framework in which MLLMs act as evaluator agents that interact directly with T2I models. Unlike previous approaches that depended on proprietary GPT-4 variants (Zhang et al., 2023) or reported poor alignment between open-source MLLMs and human judgments (Ku et al., 2024; Jiang et al., 2024b), MT2IE

demonstrates that open-source MLLMs can serve as effective T2I evaluators. To our knowledge, we are the first to explore how MLLMs can dynamically generate prompts for T2I evaluation by adapting to T2I model performance.

2. Our results show that MT2IE is able to rank the prompt-generation consistency and image aesthetics of T2I models comparably to static benchmarks, while using significantly fewer prompts and eliminating the need for prompt set curation by generating all prompts.

3. We find that MT2IE exhibits better correlation with human judgements than existing metrics on static image-text consistency benchmarks such as TIFA and DSG.

4. We demonstrate that MT2IE can also adaptively adjust prompts based on T2I model performance on previous prompts, allowing for bespoke model evaluation.

## 2 Related Work

**Alignment-based image-text consistency metrics.** Alignment-based metrics use similarity scores between image and text in the embedding space of a vision-language model (VLM). VLMs such as CLIP (Radford et al., 2021) or BLIP (Li et al., 2022) are commonly used as an automatic image-text alignment metric. The most long-standing of these metrics is CLIPScore (Hessel et al., 2021), which computes the cosine similarity of the image and text CLIP embeddings. However, many works have shown that CLIP and other VLMs treat text as a bag-of-words, failing to capture the semantics of compositional text that includes multiple objects and their attributes and spatial relations (Yuksekgonul et al., 2023; Ma et al., 2023; Thrush et al., 2022b; Kamath et al., 2023). Thus their similarity estimates are not reliable, especially when text prompts specify cardinality or compositional reasoning (Wang et al., 2023; Ma et al., 2023; Kamath et al., 2023). Despite this, many recent, impactful image and video generation methods still use CLIPScore as a prompt consistency metric (Ruiz et al., 2023; Brooks et al., 2023; Chang et al., 2023; Meng et al., 2022; Kang et al., 2023; Ho et al., 2022; Wu et al., 2023a). Similarity-based approaches have been enhanced by collecting large scale human ratings of generated images to enable finetuning of CLIP (Kirstain et al., 2023; Wu et al., 2023b). Similarly, human preference data has been used to create reward models out of alternative VLMs such as BLIP (Xu et al., 2023). Of these methods, it is unclear if any have achieved as widespread adoption as CLIPScore.

**GQA-based image-text consistency metrics.** Generated question answering (GQA) metrics for image-text consistency were introduced to overcome some of the above-described limitations. These metrics use a language model to generate questions and a VQA model to score the image using the generated questions (Cho et al., 2024; Hu et al., 2023; Changpinyo et al., 2022; Yarom et al., 2023). Interestingly, while more fine-grained, these metrics are not necessarily more reliable than alignment-based metrics (Saxon et al., 2024; Ross et al., 2024). For instance, a strong *yes*-bias is exhibited by VQA and language models (Goyal et al., 2017a; Ross et al., 2024), which are used to generate and score questions.

**Large language models as evaluators.** Following the use of LLMs for image-prompt alignment question generation, LLMs or multimodal LLMs are also being used as standalone evaluators (Lu et al., 2023; Ku et al., 2024; Zhang et al., 2023; Tan et al., 2024). Lu et al. (Lu et al., 2023) use a VLM to generate a text description of a generated image, then asks an LLM to judge the faithfulness of the image description and the image prompt. Ku et al. (Ku et al., 2024) and Zhang et al. (Zhang et al., 2023) prompt MLLMs for image-prompt faithfulness and overall image quality scores. Both works focus on using proprietary, multimodal versions of GPT-4, finding that the MLLM used has great effect on correlation with human evaluators. Notably, Ku et al. (Ku et al., 2024) find that open-sourced MLLMs are significantly worse at generated image evaluation, with Llava (Liu et al., 2023) having 90% worse human correlation compared to GPT-4v when using their evaluation methodology.

## 3 Experimental Overview

We demonstrate the effectiveness of using MLLMs as evaluator agents for assessing T2I prompt-generation consistency. In Multimodal Text-to-Image Eval (MT2IE), MLLM interacts with the T2I model by

proposing text prompts and scoring resulting generations. We show that our interactive approach enables MLLMs to effectively benchmark T2I models in two distinct settings: either by systematically generating prompts of increasing complexity or by dynamically adapting prompts based on T2I model performance. In the following sections, we focus on evaluating image-text consistency as it directly impacts the overall visual quality of generated images (Astolfi et al., 2024; Xu et al., 2023; Kirstain et al., 2023). Additionally, in Appendix A we show that MT2IE is an easily generalizable framework, capable of evaluating other aspects of T2I models such as image aesthetics. We use MT2IE to evaluate a total of 8 recent T2I models including latent diffusion model (LDM) v2.1 (Rombach et al., 2022), LDM v3 (Esser et al., 2024), LDM XL (Podell et al., 2023), LDM XL-Turbo (Sauer et al., 2025), Flux (Labs, 2024), a cascaded pixel-based diffusion model (CDM) (Raffel et al., 2020), Playground 2.5 (Li et al., 2024b) and Pixart-$\alpha$ (Chen et al., 2023). We provide an outline of our experiments and elaborate in subsequent sections.

**Experiment 1: Validation of MLLM prompt consistency scoring capabilities.**  We validate that open-sourced MLLMs are capable of scoring image-text consistency on existing benchmarks without the usage of other models or fine-tuning. Existing works that use MLLMs for end-to-end evaluation either do not use existing benchmarks or fine-tune open-sourced MLLMs for the task (Tan et al., 2024; Ku et al., 2024). Existing GQA-based benchmarks rely on a system of models for question generation and scoring; typically an LLM is used to generate questions based on a curated text prompt, the generated questions are validated by an additional LLM or question-answering model, then finally a VQA model scores the image using the generated questions (Cho et al., 2024; Yarom et al., 2023; Hu et al., 2023; Gupta & Kembhavi, 2022). We validate that MLLMs are capable of emulating GQA-based benchmarks by generating questions about a given prompt and scoring the generated questions given the corresponding image. We evaluate using TIFA160 (Hu et al., 2023), and show that a single open-sourced MLLM can recreate this benchmark and score questions with better correlation to human judgment. We detail this experiment in Section 4.

**Experiment 2: MLLMs to generate prompt consistency benchmarks.**  After validating MLLMs' abilities to generate and score prompt consistency questions in Experiment 1, we introduce MT2IE demonstrating how MLLMs can iteratively generate a full T2I benchmark and judge model text-image consistency. In particular, we show that MLLM-generated benchmarks preserve relative model rankings computed by human annotators in prior work. We also show that our method is efficient, accurately ranking models with significantly fewer prompts than existing benchmarks of handwritten prompts. We detail these experiments in Section 5.

**Experiment 3: MLLMs to generate interactive prompt consistency benchmarks.**  Beyond just generating benchmarks, we demonstrate that MT2IE can act as an evaluator agent, adapting to model-specific performance. Specifically, we show that MLLMs can adjust the complexity of prompts based on model performance on previous prompts. This enables each individual T2I model to have bespoke prompts tailored to its abilities, discussed in Section 6.

## 4 Experiment 1: Validation on Static Benchmarks & Metrics

We first validate that off-the-shelf, open-source MLLMs can effectively reproduce the results of widely used scoring approaches on benchmarks of static prompts. We explore 3 different MLLMs' abilities to generate and score questions: Molmo-7B (Deitke et al., 2024), Llama-3.2VI-11B (AI, 2024), and Llava-v1.6-34B (Liu et al., 2023). We selected only open-source models, as existing work on MLLMs as T2I evaluators focus on using proprietary models (Ku et al., 2024; Zhang et al., 2023), and we want to demonstrate that open-source MLLMs work well in our framework. We evaluate MLLMs' ability to recreate the TIFA160 benchmark (Hu et al., 2023) and use a singe MLLM to execute two different scoring approaches: (i) a generated answering (GQA) approach, where questions about a prompt are generated by an LLM and answered by a separate VQA model (*e.g.*, (Hu et al., 2023; Cho et al., 2024; Yarom et al., 2023; Changpinyo et al., 2022; Cho et al., 2023)) and (ii) a VQAScore based approach (Lin et al., 2024), which computes the likelihood of answering "Yes" to the question *"Does this figure show {text prompt}. Please answer yes or no."*.

Table 1: Human judgment correlations of existing VQA-based T2I consistency benchmarks and MT2IE, measured by Spearman's $\rho$ and Kendall's $\tau$ on TIFA160. End-to-end MLLMs have higher correlation with human judgements than existing GQA benchmarks using LLMs and VQA models.

| | $\rho$ | $\tau$ |
|---|---|---|
| **Existing GQA Benchmarks** | | |
| TIFA (Instruct-BLIP) | 46.0 | 36.0 |
| VQ$^2$A (Instruct-BLIP) | 42.6 | 35.2 |
| DSG (PaLI) | 57.1 | 45.8 |
| **MLLM Question Generation and Scoring** | | |
| Llama-3.2-11B-Vision-Instruct | 60.8 | 46.3 |
| Molmo-7b-d | 59.0 | 45.9 |
| Llava-v1.6-34b | **61.3** | **47.6** |

## 4.1 Generated Question Answering Approach

GQA-based approaches use an LLM to generate questions based on prompts then score generated questions and images using a VQA model. The performance of the VQA model being used is crucial as VQA accuracy has direct correlation with human judgment (Hu et al., 2023). For this reason, we first confirm that MLLMs have strong VQA performance on existing GQA benchmarks by using the TIFA160 LLM-generated question set. We compare the MLLMs' VQA accuracy to the 2 best VQA models from Hu et al. (2023); results are in Figure 14 (Appendix C) and show that all 3 MLLMs outperform the VQA models used in existing GQA benchmarks.

With this validation, we now investigate if a single MLLM can effectively generate and score questions given a text prompt and generated image. This constitutes a benchmark generation task, where the MLLM is responsible for creating new questions and scoring them. By consolidating these tasks to one model, this approach eliminates the resource-intensive reliance on external, often proprietary LLMs. We use all 3 MLLMs to generate and score questions about all prompts in TIFA160. The system prompts used to generate questions is shown in Appendix C.

To assess quality of the generated questions, we follow previous work and measure the correlation between the VQA accuracy (from LLM-generated questions from existing benchmarks (Cho et al., 2024; Hu et al., 2023; Gupta & Kembhavi, 2022) or MLLM-generated questions in our work) and human ratings of image-text consistency using a 1-5 Likert scale. We use images and human ratings from prior work (Cho et al., 2024) on 5 different T2I models: LDM1.1 (Rombach et al., 2022), LDM1.5 (Rombach et al., 2022), LDM2.1 (Rombach et al., 2022), minDALL-E (Saehoon et al., 2021), and VQ-Diffusion (Gu et al., 2022). We show the Spearman's $\rho$ and Kendall's $\tau$ in Table 1.

Notably, every MLLM achieves higher correlation with human annotations than existing GQA-based approaches. These results show that open-sourced MLLMs can effectively execute all tasks performed by separate models in existing GQA benchmarks. We additionally compare the GPT3-generated questions in TIFA160 and the MLLM-generated questions from Llava-v1.6-34b, the LLM with the highest human correlations. We analyze question statistics and BLEU score (Papineni et al., 2002) (a commonly used NLP metric to compare two sets of text) and find that entities in both sets of questions are very similar. Detailed results on generated question set verification are in Appendix H.

## 4.2 VQAScore Approach

VQAScore (Lin et al., 2024) computes the likelihood of the model generating "*yes*" when asked *"Does this figure show {text prompt}. Please answer yes or no."*. VQAScore has been shown to be more reliable than GQA-based approaches and alignment metrics like CLIPScore, likely due to CLIPScore's challenges with object composition (Lin et al., 2024; Tong et al., 2024; Thrush et al., 2022a; Yuksekgonul et al., 2023), and the tendency of GQA-based approaches to exhibit a "yes-bias" in question answering tasks (Ross et al., 2024). We compare the correlations with human ratings on TIFA160 between VQAScore calculated using

Table 2: Human judgment correlations of existing automatic T2I consistency metrics, including VQAScore Lin et al. (2024) computed by the CLIP-FlanT5 model and VQAScore computed by zero-shot MLLMs; measured by Spearman's $\rho$ and Kendall's $\tau$ on TIFA160.

|  | $\rho$ | $\tau$ |
|---|---|---|
| **Automatic Metrics** | | |
| BLEU-4 | 18.3 | 18.2 |
| ROUGE-L | 32.4 | 24.0 |
| METEOR | 34.0 | 27.4 |
| SPICE | 32.8 | 23.2 |
| CLIPScore (ViT-B-32) | 28.7 | 19.1 |
| **MLLM VQAScore** | | |
| VQAScore (CLIP-FlanT5, *previous SOTA*) | 66.9 | 52.8 |
| VQAScore (Llama-3.2-11B-Vision-Instruct) | 52.1 | 43.7 |
| VQAScore (Molmo-7b-d) | 49.6 | 38.4 |
| VQAScore (Llava-v1.6-34b) | **67.5** | **54.4** |

the 3 aforementioned MLLMs, VQAScore calculated with the CLIP-FlanT5 model (which was fine-tuned for SOTA VQAScore calculation (Lin et al., 2024)), and CLIPScore. We use the same human ratings as in Section 4.1; results are in Table 2. All MLLMs outperform CLIPScore in human correlation, but only Llava-v1.6-34b achieves higher correlation with human ratings than CLIP-FlanT5.

We find that **open-sourced MLLMs can replace systems of LLM and VQA models commonly used in existing QGA benchmarks.** Llava-v1.6-34b achieved the best correlation with human ratings when used to compute VQAScore as prompt consistency scores; so we use Llava-v1.6-34b with VQAScore as our MLLM evaluator for further experiments. Experiments with question generation and answering for prompt consistency scoring are in Appendix C. This validation lays the framework for our experiments, demonstrating that open-sourced MLLMs can be used for end-to-end T2I evaluation.

## 5 Experiment 2: MLLMs for Benchmark Generation

In Section 4, we showed that MLLMs are able to reproduce static benchmarks using both question-answering and prompt likelihood approaches. We now introduce our approach, **the MT2IE framework**, and show that it can be used to iteratively generate new, efficient benchmarks for image-text consistency. With each iteration, the generated prompts increase in complexity and thus challenge the T2I models in progressively more difficult settings. This involves first providing a starting prompt, which we generate in our experiments but can be passed in by a user, that we refer to as a *seed prompt*; then expanding on the seed prompt to generate increasingly more difficult and complex prompts. MT2IE-generated benchmarks are *effective* in that they match prior model rankings from benchmarks of handwritten prompts, while also being *efficient* in that they require far fewer examples than existing benchmarks. Additionally, MT2IE-generated benchmarks are dynamic, potentially addressing the dataset leakage and saturation issues of static benchmarks.

**Iterative Prompt Generation.** Our approach iterates between two stages, shown on the right side of Figure 1. First, during *iterative prompt generation*, the MLLM is given an existing prompt $T_{i-1}$ and generates a new prompt $T_i$ that builds on $T_{i-1}$ to become more difficult. $T_i$ contains an additional object, spatial relationship, or object attribute compared to $T_{i-1}$, incrementally increasing the complexity of the prompt every iteration; see Figure 2 for an example. The seed prompt $T_0$ can be user-written or generated. The T2I model generates an image $I_i$ based on the prompt $T_i$. Next, during *image scoring*, the MLLM scores $I_i$ based on its faithfulness to prompt $T_i$. A full run consists of $N$ iterations of both iterative prompt generation and image scoring stages, with each iteration yielding a score $s_i$. We perform 756 full runs, with $N = 5$ iterations per-run, resulting in a total of 3,780 scores from all runs. For these runs we evaluate a subset of 4 T2I models: CDM, LDM v2.1, LDM XL and LDM XL-Turbo. Our framework is flexible and supports any scoring metric during the image scoring stage; in this paper we use VQAScore (Lin et al., 2024), as it achieved the highest correlations with human judgment (see Experiment 1 in Section 4.2).

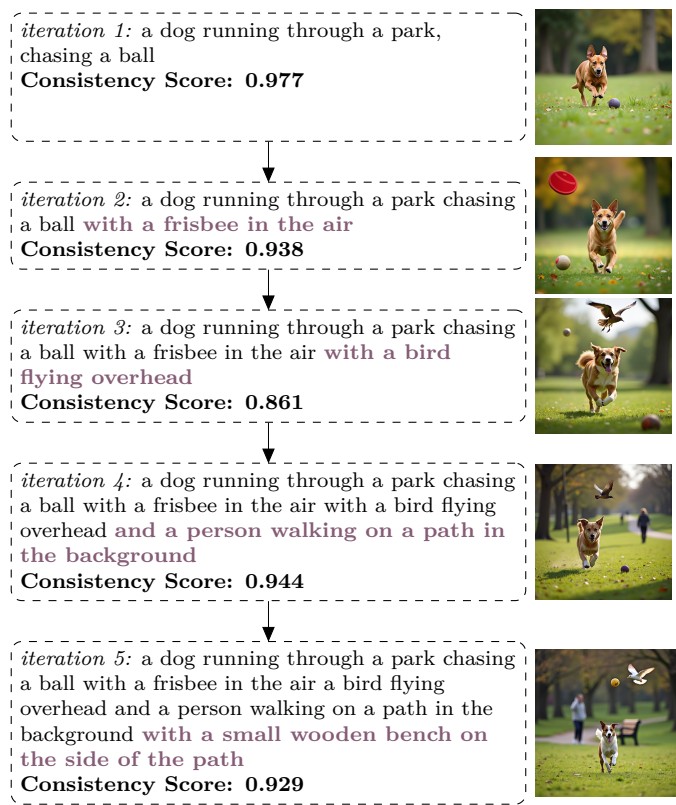

Figure 2: An example our method MT2IE generating progressively difficult prompts, with additions **shown in purple**. Corresponding generated images and scores are also shown. Prompts become more complex while maintaining linguistic structure.

Figure 3a shows average scores for each prompt iteration. We find that scores for all models decrease as prompt iterations increase, indicating that MT2IE effectively generates prompts that are progressively more challenging. To support this, we also analyze the change in linguistic properties of prompts across iterations in Figure 3b; we use the Flesh-Kincaid score (Flesch, 1948) to compute semantic difficulty. As MT2IE-generated prompts get more difficult, they still maintain coherence and grammaticality. In general, prompts increase in their word count and semantic difficulty as prompt iterations occur. See an example progression of a 5-iteration run in Figure 2.

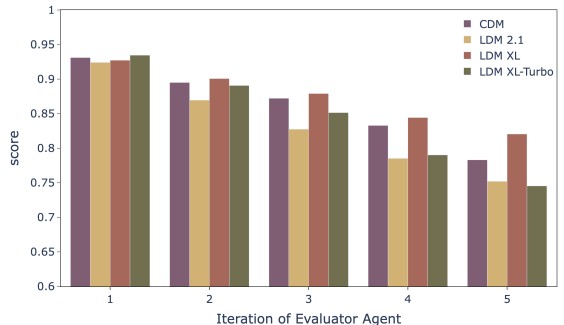

(a) The progression of model scores as the prompts become increasingly more difficult.

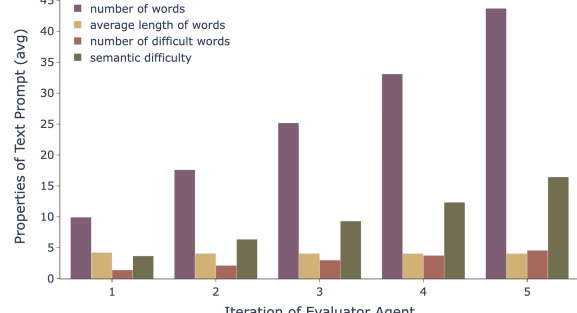

(b) Shifts in various linguistic properties, averaged across all prompts in a given iteration.

Figure 3: Results for MT2IE, where the MLLM iteratively generates progressively more difficult prompts over 5 iterations.

Table 3: Rank correlations between T2I model rankings on full GenAIBench (1600 prompts) versus only 20 prompts using our method versus all other evaluation methods (also see Figure 4). MT2IE produces notably high correlations rankings.

|  | VQAScore | CLIPscore | VIEScore | MT2IE (Ours) |
|---|---|---|---|---|
| Kendall's $\tau$ | $-0.6428$ | $0.3571$ | $0.2857$ | $0.8571$ |

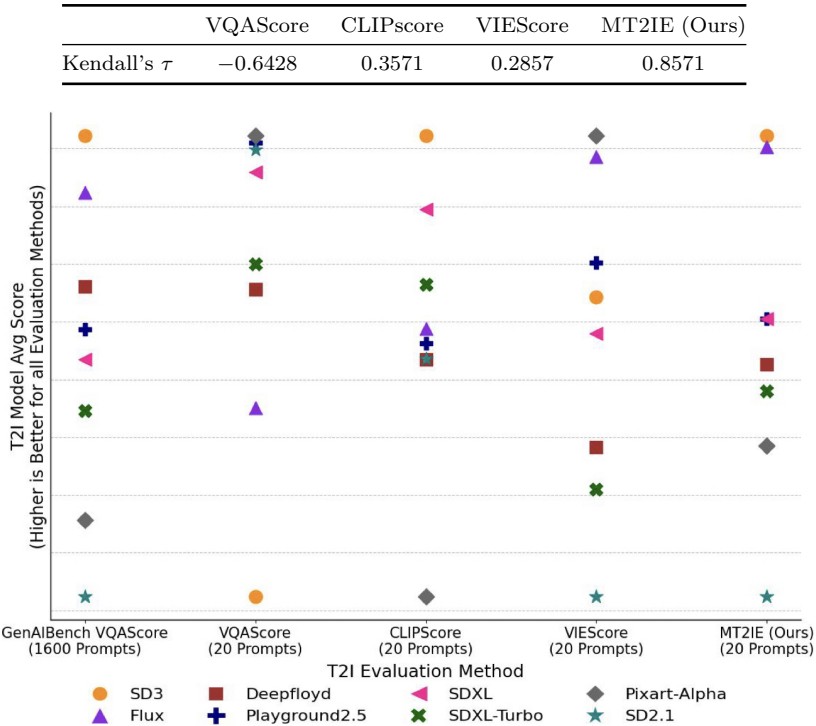

Figure 4: Model rankings produced by MT2IE closely match GenAIBench's ranking, while only using 20 generated prompts. Other methods produce mismatched model rankings when using 20 sampled GenAIBench prompts.

**More Efficient Evaluation using our generated prompts.** After establishing that MT2IE generates progressively challenging prompt consistency benchmarks and achieves higher correlations in scoring generations, we next investigate how efficient MT2IE-generated benchmarks are compared to static benchmarks. Concretely, we ask: **can we achieve the same evaluation results, *i.e.*, same model rankings, while requiring fewer text prompts?** We start by generating 4 seed prompts across 4 topic categories that encompass COCO's captions: household scenes (foods, household items, or furniture), descriptions of people, scenes with animals, and location descriptions Lin et al. (2014). Each seed prompt is moderately complex (2-3 compositional concepts per prompt) and undergoes 4 difficulty progression iterations ($T_0 \rightarrow T_4$), resulting in 20 total evaluation prompts in the generated benchmark (4 seeds $\times$ 5 prompts). Empirically we found that 5 prompts, resulting from 4 iterations, provided a sufficient amount of prompt difficulty (see Appendix G for ablations on number of iterations). We refer to this as MT2IE$_{bench20}$.

We rank the 8 T2I models listed in Section 3 by (i) average score on 20 generated prompts from MT2IE$_{bench20}$ and (ii) VQAScore evaluated on 1600 human-written prompts from GenAIBench (Li et al., 2024a). Next, for a direct comparison we randomly sample 20 prompts from GenAIBench, ensuring uniform sampling across all prompt categories, and refer to this as GenAIBench$_{20}$. We rank the T2I models on GenAIBench$_{20}$ by (iii) VQAScore (iv) CLIPScore (v) VIEScore. We run MT2IE-benchmark generation over 5 random seeds to ensure results are statistically significant (model score mean and standard deviations in Figure 7 and Appendix C Figure 10). Because variance across 5 seeds was extremely high for all metrics ran on GenAIBench$_{20}$, all GenAIBench$_{20}$ results are reported over 10 seeds. The relative rankings and scores for each evaluation method are shown in Figure 4.

We find that MT2IE, generating and scoring only 20 prompts per ranking, manages to closely match the T2I model ranking from the full, 1600-prompt GenAIBench benchmark. This indicates that MT2IE matches the T2I evaluation effectiveness of leading large-scale benchmarks while utilizing 20 times fewer prompts. Additionally, MT2IE-generated prompts eliminate the need for costly human-written prompt

sets for benchmarking. All other evaluation methods tested failed to reliably match model rankings when applied to different random samples of 20 prompts from GenAIBench. For instance, VQAScore's ranking on GenAIBench$_{20}$ shows LDM v2.1 outperforming Flux, which is inconsistent with every other benchmark (Labs, 2024). These results show that simply randomly sampling a subset of prompts from existing datasets is not enough; our iterative approach generates strong examples that are able to truly probe performance.

## 6 Experiment 3: MLLMs as Adaptive Evaluator Agents

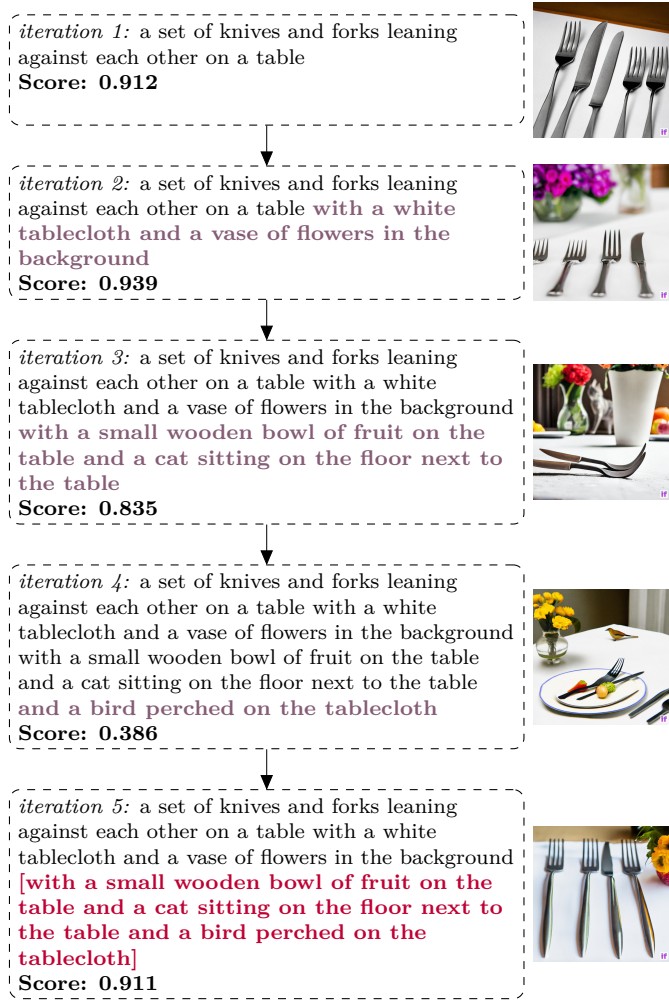

Figure 5: Example run with MT2IE's Adaptive Evaluation, with generated images and alignment scores to illustrate the evaluation process. **Portions in purple** were added to the prompt from the previous iteration; **portions in red** were removed. Generations have high scores for iterations 1-3, so the prompts get progressively more difficult. When the score drops significantly at iteration 4, MT2IE reacts and simplifies the prompt.

We demonstrated that the MT2IE framework can iteratively generate prompts of increasing complexity, providing more efficient benchmarks compared to traditional static prompt sets. In this section, we move a step further in benchmark generation and ask: **Can MT2IE customize evaluations for individual T2I models based on their performance history?** Individualized evaluation provides a nuanced understanding of differences between models and can facilitate the automatic discovery of model failure points. We explore adding an adaptive aspect to MT2IE, where the MLLM interacts with the T2I model by adjusting prompts based on the generated images seen from previous iterations.

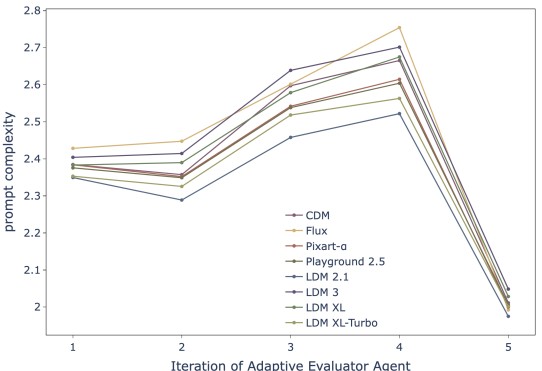 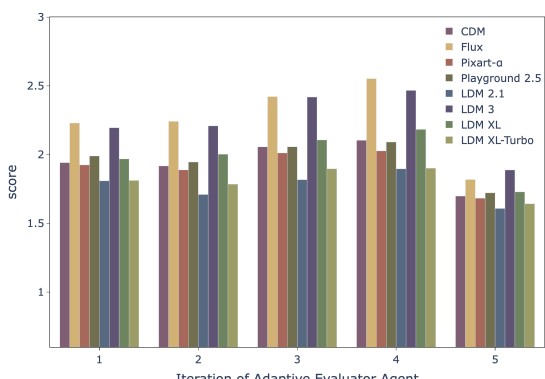

(a) Prompt syntactic complexity, averaged across all prompts at each iteration. While most models get progressively more complex prompts from the first iteration, weaker models like LDM 2.1 struggle. All models hit a performance wall at the 4th iteration.

(b) Using MT2IE, we show how the scores for 8 T2I models change in our adaptive framework.

Figure 6: Results for MT2IE, where the MLLM iteratively generates prompts *with feedback* on individual model scores over 5 iterations. The difficulty of the generated prompt is dependent on the score from the previous iteration; description in Section 6.

**Adaptive Prompt Generation.** As previously described, MT2IE iteratively expands upon prompts and scores corresponding image generations. In this adaptive setting, the MLLM additionally incorporates feedback about how well the T2I model performed on the previous iteration, via the score (Figure 1, right). This approach enables the MLLM to adaptively generate more challenging prompts when the T2I model performs well and simpler prompts when the model struggles. This dynamic adjustment is desirable because it enables discovery of failure patterns in strong models while avoiding uninformative results from excessively difficult prompts for weak models.

In this setting, the MLLM is instructed to update prompt $T_i$ based on the prompt consistency score of the previous prompt $T_{i-1}$ according to:

- score $0.0 - 0.2$: complexity halved
- score $0.2 - 0.4$: complexity reduced by 1-2 terms
- score $0.4 - 0.6$: prompt rephrased, same complexity
- score $0.6 - 0.8$: complexity increased by 1-2 terms
- score $0.8 - 1.0$: complexity increased by 2+ terms

With this system, T2I models that perform very poorly (score $\leq 0.2$) on prompt $T_i$ will have a significantly simpler next prompt $T_{i+1}$. Conversely, T2I models that are performing very well (score $\geq 0.8$) will have a significantly more difficult next prompt. To compute the final score after all $N$ iterations, each prompt's score is weighted by its difficulty, ensuring greater rewards for models excelling at harder prompts compared to those scoring highly on easier ones. We show how the prompt complexity shifts across iterations for each model in Figure 6a. Older, less advanced T2I models, *e.g.*, LDM 2.1, show reduced prompt complexity almost immediately. Recent, high-performing T2I models, *e.g.*, Flux, receive the most difficult prompts with MT2IE. An example of an adaptive run for CDM (Raffel et al., 2020) is shown in Figure 5.

**Adaptive Prompting Preserves T2I Model Rankings.** Following our approach in Section 5, we use MT2IE to rank the same 8 T2I models on 20 adaptively generated prompts across 5 random seeds. We compare these rankings to GenAIBench's rankings obtained from both its complete 1,600-prompt benchmark and a sampled 20-prompt subset, calculated using VQAScore from the CLIP-FlanT5 model. For the 20-prompt downsample of GenAIBench we report results over 10 random seeds, sampling representatively across all prompt categories, to ensure fairness. Because the adaptively generated prompts are custom to each model and therefore not directly comparable, we first standardize the scores to reflect prompt

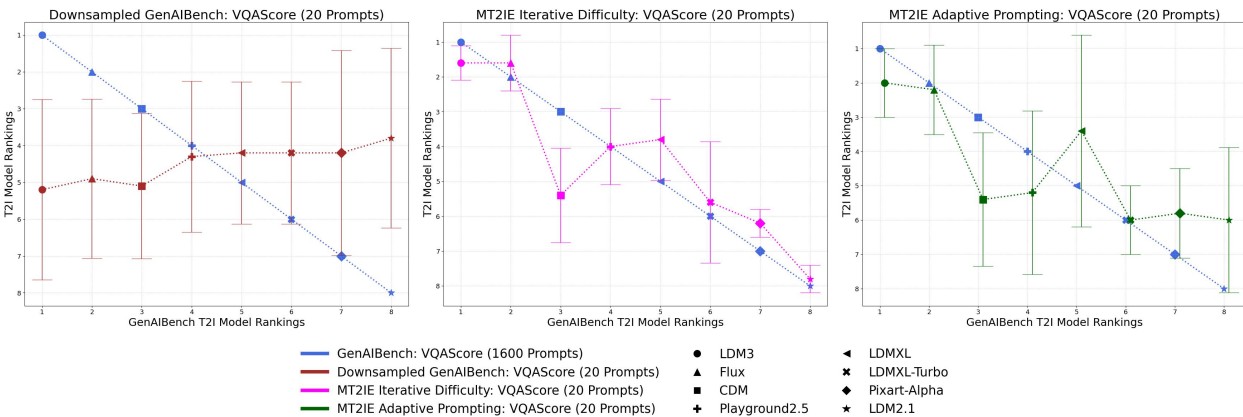

Figure 7: Our MT2IE framework produces T2I model rankings consistent with VQAScore on GenAIBench's 1600 hand-written prompt benchmark (**blue**) with just 20 generated prompts in both the progressive difficulty setting (**pink**) and the model adaptive setting (**green**). When we downsample GenAIBench to 20 prompts (**red**), the rankings are noisy and uninformative, showing the effectiveness of our framework while using 80× less evaluation examples. Average rank (1=highest, 8=lowest) and rank variance over all seeds are plotted.

difficulty and then rank models with standardized scores. We use *syntactic complexity* to approximate prompt difficulty, computed using Yngve score which is a measure of the depth of a syntactic parse tree (Yngve, 1960). Appendix D includes our reasoning for using Yngve score as well as a correlation analysis of multiple NLP language complexity metrics that could serve as potential prompt difficulty scores, including Flesch–Kincaid grade level (Flesch, 1948) and perplexity (Jelinek et al., 1977). For each prompt-image pair, the initial consistency score is multiplied the prompt difficulty score to obtain the final score.

We show the average rank of each T2I model over all run seeds in Figure 7. Model rankings produced by MT2IE with adaptive prompting have higher variance across seeds than MT2IE with iterative difficulty, but still generally match the ranking from GenAIBench's set of 1600 handwritten prompts, even when each individual model is evaluated on its bespoke prompt set. We again demonstrate the efficiency of our approach, requiring only 20 T2I generations to match evaluation results from an 80× larger, handwritten prompt benchmark.

# 7 Conclusion

Our work leverages the power of multimodal LLMs (MLLMs) as evaluator agents for text-to-image (T2I) model evaluation, and introduces the MT2IE framework. Through MT2IE, we show that MLLMs are able to generate dynamic benchmarks of prompts. These benchmarks are effective for model evaluation, matching existing human-annotated rankings, and also highly efficient by requiring only a fraction of the examples. We also demonstrate that incorporating model-specific feedback to craft prompts enables the assessment of individual model's abilities. Our MT2IE framework may be easily used with any off-the-shelf MLLM and incurs no computational cost on top of regular inference; MLLM system prompts and details on ease of adoption can be found in Appendices B, C, and F.

Robust metrics and benchmarks are crucial to understanding and advancing T2I models. The current paradigm of T2I model evaluation relies heavily on legacy, static benchmarks (*e.g.*, COCO (Lin et al., 2014), PartiPrompts (Yu et al., 2022)), which can be costly to create and quickly saturate (Ott et al., 2022; Gupta et al., 2024). MT2IE's effectiveness shows that MLLMs can be used in T2I evaluation to generate robust, efficient prompts to benchmark generation consistency and quality. We hope the community will build on these results to further develop dynamic and interactive evaluation frameworks, bridging the gap towards agentic-based evaluations.

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

# Appendix

## A    MT2IE For Aesthetic Quality Evaluations

To demonstrate MT2IE's ability to generalize beyond evaluating image-prompt consistency, we use it to evaluate image aesthetic quality and compare its aesthetic quality judgments to a state-of-the-art aesthetics scoring model. MT2IE is easily adaptable to be used to evaluate other aspects of generated images by simply using a different set of MLLM system prompts; for aesthetic quality evaluation we use the system prompts in Figure 17. All other aspects of MT2IE are kept the same, and we use the 800 generated prompts and images from Section 5. We compare aesthetic scores produced by MT2IE and the aesthetics predictor of LAION (2022), a popular aesthetics-scoring model (Kirstain et al., 2023; Wu et al., 2023b; Xu et al., 2023). MT2IE 's ranking of 8 T2I models by average aesthetic score closely matches the pre-trained aesthetics-scoring model's ranking, with a *Kendall's $\tau$ of* 0.691. Per-image aesthetic scores of 800 images are also positively correlated with a *Kendall's $\tau$ of* 0.538. These results, in tandem with our main paper experiments, show that MT2IE can effectively evaluate aesthetic quality and prompt consistency. Additionally, we demonstrate that MT2IE can be easily repurposed to evaluate axes other than prompt consistency by rewriting the MLLM prompt used for judgement.

```
 You are a professional image judge.  You will give a score between 0 and 10 to the
given images about how visually pleasing they are.
Here are things to consider when scoring the images:
How clear and sharp is the image?
Is anything in the image malformed?
Are the colors and figures in the image pleasing and coherent?

A score of 0 indicates that the image is very blurry, only has malformed objects,
and unpleasant colors.
A score of 10 indicates that the image is clear, all objects are nicely depicted, and
the colors are pleasant.
```
```
Score this image between 0 and 10.  Only state the score and nothing else.  (image)
```

Figure 8: Aesthetic quality evaluation system prompt (top) and user prompt given per-image evaluation (bottom).

## B    Ease of Adoption

The computational cost of one evaluation run with MT2IE is 40 inference passes of the utilized MLLM: 20 prompt generations with one prompt-image judgment each. Exact time and memory costs depend on MLLM size. However MT2IE greatly reduces the computational cost of generating images, as it is capable of matching T2I model rankings when evaluating on 20x less image-text pairs, which offsets the MLLM computation costs. MLLM system prompts for MT2IE are provided in Appendix F, C, and can be easily used with any off-the-shelf MLLM.

## C    VQA Accuracy Scoring Results and MLLM System Prompts

In Section 4, we validate that MLLMs are able to effectively score T2I models. We have two scoring paradigms: VQA Accuracy (question generation and answering) and VQAScore. VQA Accuracy requires generating multiple choice questions given a description. In the main paper, we only discuss VQAScore prompt alignment scoring as it acheived the highest correlation with human judgment in experiments in Section 4. We therefore provide more details on the VQA Accuracy process below.

We provide the prompt given to each MLLM for question generation and validation (we only use questions for which the answer to the validation prompt is "yes"). Figure 11 shows Llama-3.2-11B-Vision Instruction, Figure 12 shows Molmo-7b-d and Figure 13 shows Llava-v1.6-34b. We also provide experimental results for MT2IE with both prompt alignment scoring paradigms discussed in Figures 9 and 10.

MT2IE with both VQA Accuracy and VQAScore scoring are able to match GenAIBench's T2I model rankings while using 80x less prompts. For MT2IE with progressively harder prompts, T2I model scores from both VQA Accuracy and VQAScore scoring are extremely consistent if not exactly matching, showing that MLLM's question generation and VQA abilities are on-par with the SOTA VQAScore metric.

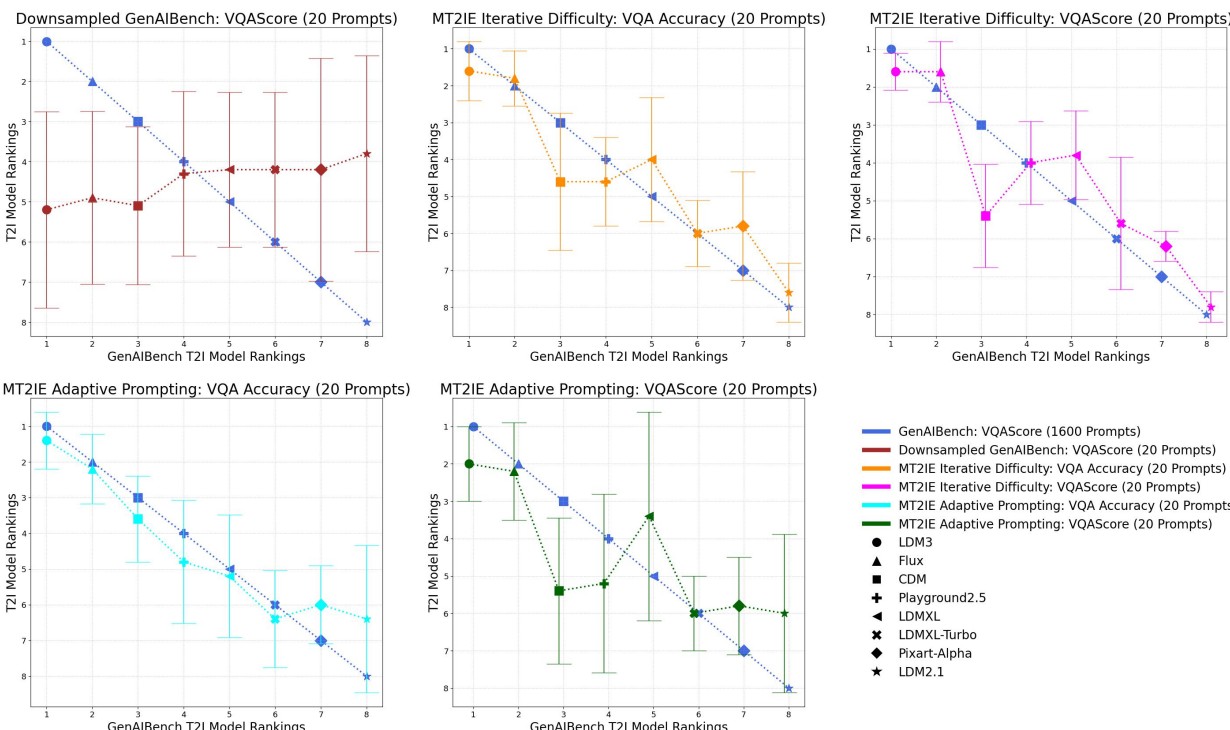

Figure 9: Average rank of 8 T2I models (shown on x axis where 1=highest scoring, 8=lowest scoring) evaluated by MT2IEwith adaptive prompting and progressively difficult prompt generation, where each prompt generation method is run with both question generation and answering (VQA Accuracy) or VQAScore for prompt alignment scoring. All MT2IEsettings evaluate T2I models over 20 generated prompts, and we compare to ranks from GenAIBench with CLIP-FlanT5 VQAScore. MT2IErankings generally match GenAIBench's T2I model rankings, while the Downsampled GenAIBench model rankings are uninformative as all models have roughly the same mean rank over all seeds and high variation. See Figure 4 for details on GenAIBench and Downsampled GenAIBench.

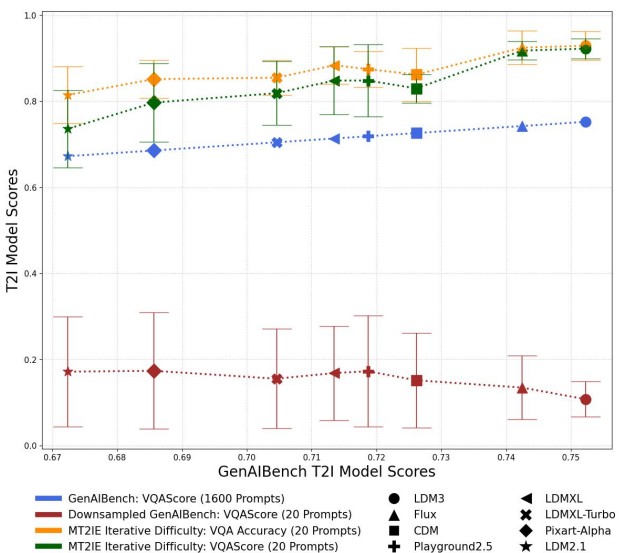

Figure 10: We evaluate 8 T2I models, comparing MT2IEwith progressively difficult prompt generation, and scoring prompt alignment with both question generation and answering (VQA Accuracy) or VQAScore, to other evaluation approaches. Scores from MT2IE are obtained from just 20 prompts that are both generated and scored by an MLLM. Scores from GenAIBench are obtained by generating and scoring 1600 human-written prompts. We find consistent T2I model rankings and scores between MT2IE, using only 20 prompts, and the full GenAIBench with both VQA Accuracy and VQAScore prompt alignment scoring methods (discussed in 4). Just downsampling GenAIBench to 20 prompts breaks the relative model ranking.

```
 Given an image description, generate multiple-choice questions that check if an
image depicts everything in the image description.
You will not be given an image, just generate multiple-choice questions for the given
image description and write nothing else.

First list the objects, attributes, spatial relations, and actions in the image
description.  Then write one questions for each of the stated items.
Write one question checking if each object is in the image.  For any object
attributes, write a question checking if the object in the image has the attribute.
For any spatial relations or actions, write a question checking if the spatial
relation or action is shown in the image.

For each multiple-choice question, list answer choices and state the correct answer.
First write "Q:" and then state the question and nothing else.
Then on the next line write "Choices:  " and then list the answer choices all on one
line.
Then on the next line write "A: " and then state which answer out of the choices is
correct.
All answer choices should be relevant to the question and short.
```
```
The only information you have is this image description:  (prompt)
Can this question be answered with only the image description?
Question:  (question)
Answer yes or no and state nothing else.
```

Figure 11: For Llama-3.2-11B-Vision-Instruct, we provide the prompt for the question generation (top) and question validation (bottom).

```
 Given an image description, generate multiple-choice questions that check if an image matches the image description.
You will not be given an image, just generate multiple-choice questions for the given image description and write
nothing else.
For each multiple-choice question, generate answer choices and the correct answer.
Write one question for each object, attribute, and spatial relation in the image description.
Do not write questions about attributes of objects not mentioned in the description.  Do not list more than 4 answer
choices.

Generate questions, answer choices, and answers formatted like these examples:
Image description:  a drawing of a red crab
Q: Is there a crab in the image?
Choices:  yes, no
A: yes
Q: What animal is this?
Choices:  lobster, fish, crab, eel
A: crab
Q: Is this a drawing?
Choices:  yes, no
A: yes
Q: What color is the crab?
Choices:  grey, blue, red, purple
A: red

Image description:  A cheese grater with a red handle sitting on a white countertop next to a yellow cutting board
and a green apple
Q: Is there a cheese grater?
Choices:  yes, no
A: yes
Q: What kitchen tool is on the counter?
Choices:  blender, coffee maker, cheese grater, mixer
A: cheese grater
Q: What color is the handle of the cheese grater?
Choices:  green, silver, white, red
A: red
Q: What color is the countertop?
Choices:  black, white, marble, grey
A: white
Q: Is there a cutting board?
Choices:  yes, no
A: yes
Q: What color is the cutting board?
Choices:  white, yellow, blue, brown
A: yellow
Q: Is there an apple?
Choices:  yes, no
A: yes
Q: What color is the apple?
Choices:  yellow, red, green
A: green

Image description:  laundry detergent
Q: Is there laundry detergent in the image?
Choices:  yes, no
A: yes
Image description:  a tropical rainforest with a waterfall cascading down a cliff a group of monkeys swinging from the
trees and a large colorful parrot perched on a branch
Q: Is there a tropical rainforest?
Choices:  yes, no
A: yes
Q: Is there a waterfall cascading down a cliff?
Choices:  yes, no
A: yes
Q: Are there a group of monkeys in the image?
Choices:  yes, no
A: yes
Q: What are the monkeys doing?
Choices:  eating, swinging from trees, swimming, sleeping
A: swinging from trees
Q: Is there a parrot in the image?
Choices:  yes, no
A: yes
Q: Where is the parrot in the image?
Choices:  flying overhead, perched on a branch, standing on a rock, flying above the water
A: perched on a branch
```

```
 Given this image caption:  (prompt)
Does this question ask about information explicitly included in the caption:  (question)
Answer yes or no and state nothing else.
```

Figure 12: For Molmo-7b-d, we provide the prompt for the question generation (top box) and question validation (bottom box).

```
 Given an image description, generate multiple-choice questions that check if an image matches the image description.
You will not be given an image, just generate multiple-choice questions for the given image description and write
nothing else.
For each multiple-choice question, generate answer choices and the correct answer.
Write one question for each object, attribute, and spatial relation in the image description.
Do not write questions about attributes of objects not mentioned in the description.  Do not list more than 4 answer
choices.

Generate questions, answer choices, and answers formatted like these examples:
Image description:  a drawing of a red crab
Q: Is there a crab in the image?
Choices:  yes, no
A: yes
Q: What animal is this?
Choices:  lobster, fish, crab, eel
A: crab
Q: Is this a drawing?
Choices:  yes, no
A: yes
Q: What color is the crab?
Choices:  grey, blue, red, purple
A: red

Image description:  A cheese grater with a red handle sitting on a white countertop next to a yellow cutting board
and a green apple
Q: Is there a cheese grater?
Choices:  yes, no
A: yes
Q: What kitchen tool is on the counter?
Choices:  blender, coffee maker, cheese grater, mixer
A: cheese grater
Q: What color is the handle of the cheese grater?
Choices:  green, silver, white, red
A: red
Q: What color is the countertop?
Choices:  black, white, marble, grey
A: white
Q: Is there a cutting board?
Choices:  yes, no
A: yes
Q: What color is the cutting board?
Choices:  white, yellow, blue, brown
A: yellow
Q: Is there an apple?
Choices:  yes, no
A: yes
Q: What color is the apple?
Choices:  yellow, red, green
A: green

Image description:  laundry detergent
Q: Is there laundry detergent in the image?
Choices:  yes, no
A: yes
```

Figure 13: For Llava-v1.6-34b, we provide the prompt for the question generation (top) and question validation (bottom).

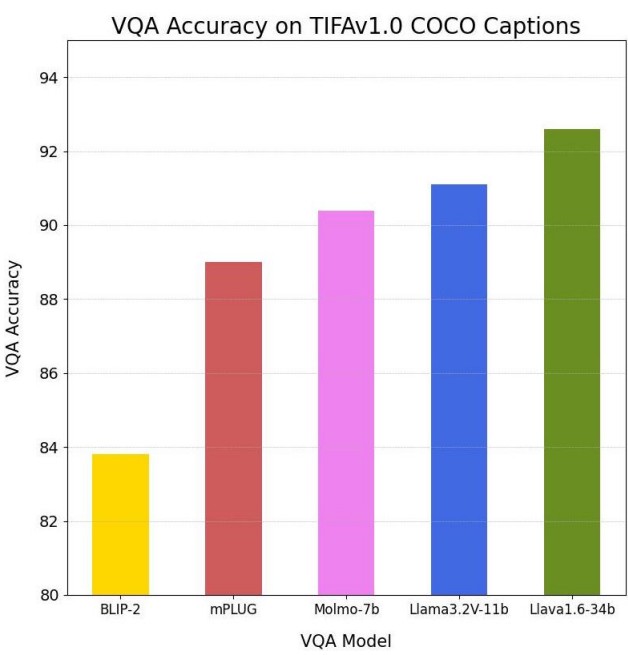

Figure 14: All MLLMs obtain higher VQA accuracy than VQA models used as image judges in previous work. Results are on the 2k COCO image-caption pairs in TIFA160, as done in Hu et al. (2023).

# D  Analysis of Prompt Difficulty Metrics

For MT2IEwith Adaptive Prompting, discussed in Section 6, we multiply prompt alignment scores with a prompt difficulty metric, in order to standardize prompt alignment scores since all models are no longer evaluated on the same prompts. An ideal prompt difficulty metric reflects how difficult it is for a T2I model to generate an image faithful to the prompt, and can weight the final prompt alignment score appropriately: scaling difficult prompts to higher values and easy prompts to lower values.

However to our knowledge, there is no existing work exploring metrics for T2I prompt difficulty. So we considered various NLP metrics for language complexity as prompt difficulty proxy metrics, including: word count, syllable count, Yngve score (Yngve, 1960) (a measure of the depth of a syntactic parse tree), Flesch–Kincaid scores (Flesch, 1948), and perplexity (Jelinek et al., 1977). *We did preliminary analysis on a small set of MLLM-generated prompts and VQAScores and found that Yngve score was the metric most correlated with VQAScore,* so we used Yngve score as the prompt difficulty weight in Section 6's experiments. After conducting all our experiments, we now have 20,640 (MLLM-generated) prompts along with their corresponding T2I VQAScores to analyze in order to identify a strong potential metric for prompt difficulty.

Table 4 shows correlations between various metrics calculated on prompts and T2I VQAScores for those same prompts. Interestingly, the relativey simple average word length statistic performs the best, outperforming semantic complexity metrics like Flesh-Kincaid grade level. We found perplexity to be a highly unstable metric, showing extreme variation across relatively similar prompts, often differing by thousands. Prepending "There's an image of a" to the prompts stabilized perplexity values, which we do before calculating perplexity for all results. Effectively measuring prompt difficulty is an interesting area of future work that we believe is very useful to the field of T2I evaluation.

Table 4: Prompt metric correlations of 20,640 T2I model VQAScores of prompts and their generated images. The higher the correlation, the more viable the prompt metric is as a prompt difficulty metrics. Perplexity was calculated with Llama2, and we pre-pended "There's an image of a" to the prompt to stabilize perplexity values.

| Prompt Metric | Kendall's $\tau$ | Spearman's $\rho$ |
|---|---|---|
| Flesch-Kincaid Grad Level | $-0.1368$ | $-0.1011$ |
| Word Count | $-0.1511$ | $-0.1188$ |
| Syllable Count | $-0.1437$ | $-0.1076$ |
| Average Syllables per Word | $0.0476$ | $0.1022$ |
| Average Word Length | $0.1025$ | $0.1739$ |
| Perplexity (Llama2) | $0.0574$ | $0.0407$ |
| Yngve Score | $-0.0970$ | $-0.0965$ |

# E MT2IE With Various Multimodal LLMs

We run MT2IE using the Iterative Difficulty setting described in Section 5, incorporating two additional MLLMs: llava-v1.6-vicuna-13b-hf and llava-v1.6-vicuna-7b-hf. These models have 13 billion and 7 billion parameters, respectively, and are smaller variants of the same Llava architecture as the MLLM used in our main experiments, Llava-v1.6-34b.

As shown in Figure 15 and Table 5, MT2IE achieves higher rank correlations with the 1600-prompt GenAIBench benchmark than other evaluation methods, even when using smaller MLLMs. As the size of MLLMs increases, rank correlations become stronger, leading to better alignment of T2I model rankings when using larger, more capable MLLMs. Variance in T2I model scores across seeds also decrease when MT2IE is used with larger MLLMs. These results are consistent with previous findings on MLLMs, where larger models achieve better and more reliable visual understanding and reasoning performance.

Table 5: Rank correlations between T2I model rankings on full GenAIBench (1600 prompts) versus only 20 prompts using MT2IE with various MLLMs (also see Figure 15). MT2IE produces higher rank correlations than existing evaluation methods even when using the smallest MLLM, however rank correlation scales with MLLM size.

|  | VQAScore | CLIPscore | VIEScore | MT2IE (32B) | MT2IE (13B) | MT2IE (7B) |
|---|---|---|---|---|---|---|
| Kendall's $\tau$ | $-0.6428$ | $0.3571$ | $0.2857$ | $0.8571$ | $0.7857$ | $0.5714$ |

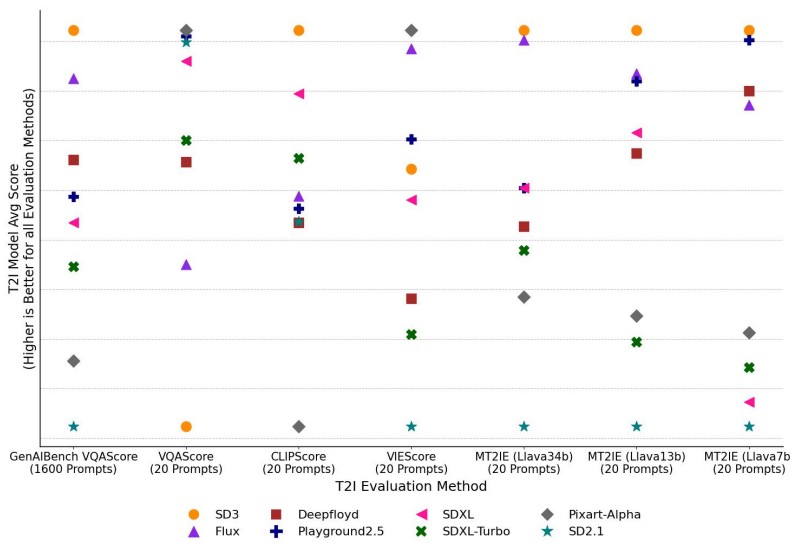

Figure 15: Model rankings produced by MT2IE used with 3 different MLLMs of varying sizes. As expected, the largest MLLM best matches GenAIBench's ranking, however the smaller MLLMs rankings still have higher correlation with GenAIBench over existing methods.

## F  MLLM System Prompts for Prompt Generation

**MT2IE Iterative Difficulty Prompt Generation (Section 5).**   For prompt generation in Section 5, we use the system prompt shown in Figure 16 and run generation with temperature 0.3 and no top-k sampling.

```
 You are responsible for rewriting prompts for a computer program that generates
images.
You will be given an existing prompt that you must add complexity to by adding one
term that is an additional object, a new spatial relationship, or new attribute.
Do not add subjective descriptors to the prompt, it should be as factual as possible.
Do not use fluffy, poetic language, or any words beyond the elementary school level.
Every time you will be givng an existing prompt, write "Prompt:" and write the new
prompt, do not state anything else.

Existing prompt:  "a serene mountain lake with a small wooden dock and a small wooden
cabin on the shore"
Prompt:  a serene mountain lake with a small wooden dock and a small wooden cabin on
the shore, with birds in the sky

Existing prompt:  "elderly man wearing a backpack and a hat walking with a cane and a
dog on a leash"
Prompt:  elderly man wearing a backpack and a hat walking with a cane and a big white
dog on a leash

Existing prompt:  "a green fish hiding in a coral reef next to a crab"
Prompt:  a green fish hiding in a coral reef next to a crab and a purple octopus
Existing prompt:  (previous prompt)
```

Figure 16: System prompt for Llava-v1.6-34b when generating iteratively more difficult prompts.

**MT2IE: Aesthetics Scoring (Section ??).**   For aesthetic quality evaluation we use the system prompts in Figure 17.x

```
 You are a professional image judge.  You will give a score between 0 and 10 to the
given images about how visually pleasing they are.
Here are things to consider when scoring the images:
How clear and sharp is the image?
Is anything in the image malformed?
Are the colors and figures in the image pleasing and coherent?

A score of 0 indicates that the image is very blurry, only has malformed objects,
and unpleasant colors.
A score of 10 indicates that the image is clear, all objects are nicely depicted, and
the colors are pleasant.
```
```
Score this image between 0 and 10.  Only state the score and nothing else.  (image)
```

Figure 17: Aesthetic quality evaluation system prompt (top) and user prompt given per-image evaluation (bottom).

**MT2IE Adaptive Prompting Prompt Generation (Section 6).**  For adaptive prompt generation in Section 6, we use the system prompts shown in Figures 18, 19, 20 and run generation with temperature 0.3 and no top-k sampling.

**Score** [0.0, 0.2]:
You are responsible for rewriting prompts for a computer program that generates
images.
You will be given an existing prompt that you must simplify so that there are half
the amount of terms.
Remove objects, attributes, or spatial relationships in the existing prompt and
rewrite the prompt so that the new prompt makes sense but only describes half the
amount of concepts.
If the existing prompt is too simple to reduce, just rewrite the prompt to be
clearer.
Every time you will be givng an existing prompt, write "Prompt:" and write the new
prompt, do not state anything else.

Existing prompt: "a serene mountain lake with a small wooden dock a few birds flying
overhead and a small wooden cabin on the shore"
Prompt: a mountain lake with a wooden dock

Existing prompt: "elderly man walking with a cane a dog on a leash a backpack on
his back a hat on his head and a pair of sunglasses on his face" Prompt: an elderly
man walking with a cane and a dog

Existing prompt: "green fish swimming in a coral reef next to a purple octopus"
Prompt: a green fish in a coral reef

**Score** (0.2, 0.4]:
You are responsible for rewriting prompts for a computer program that generates
images.
You will be given an existing prompt that you must simplify by removing two terms.
The terms to remove are objects, attributes, or spatial relationships in the existing
prompt.
If there are less than two terms in the existing prompt, just rewrite the prompt to
be simpler.
Every time you will be givng an existing prompt, write "Prompt:" and write the new
prompt, do not state anything else.

Existing prompt: "a serene mountain lake with a small wooden dock a few birds flying
overhead and a small wooden cabin on the shore"
Prompt: a mountain lake with a small wooden dock and a few birds

Existing prompt: "elderly man walking with a cane a dog on a leash a backpack on
his back a hat on his head and a pair of sunglasses on his face"
Prompt: elderly man with a cane wearing a backpack and a dog on a leash

Existing prompt: "a green, striped fish hiding in a coral reef next to a crab"
Prompt: a green fish in a coral reef

Figure 18: System prompts for Llava-v1.6-34b when generating adaptive prompts, for each of the score bins listed.

---

**Score** $(0.4, 0.6)$**:**
You are responsible for rewriting prompts for a computer program that generates
images.
You will be given an existing prompt that you must rewrite to make more clear and
simple.
Rewrite the prompt with simpler words, clearer phrasing, and add punctuation if
needed.  The meaning of the new and existing prompts must be the same.
Every time you will be givng an existing prompt, write "Prompt:" and write the new
prompt, do not state anything else.

Existing prompt:  "a serene mountain lake with a small wooden dock a few birds flying
overhead and a small wooden cabin on the shore"
Prompt:  a lake on a mountain with a small dock, birds flying in the sky, and a small
cabin on the shore

Existing prompt:  "elderly man walking with a cane a dog on a leash a backpack on
his back a hat on his head and a pair of sunglasses on his face"
Prompt:  an old man wearing a backpack, a hat, and sunglasses walking with a cane and
his dog on a leash

Existing prompt:  "a group of seals jumping out of the water in a large body of
water with a boat nearby and a person on the boat taking a picture while a seagull
is flying overhead"
Prompt:  seals jumping out of a large body of water, near a boat with a person taking
a picture onboard and a seagull flying overhead

Existing prompt:  "a city with crowded streets and skyscrapers with a mountain in
the background and traffic in the foreground and a large stadium in the distance"
Prompt:  a city with busy streets and tall buildings and a mountain in the background
and cars in the foreground, with a large stadium in the distance

---

**Score** $[0.6, 0.8)$**:**
You are responsible for rewriting prompts for a computer program that generates
images.
You will be given an existing prompt that you must add complexity to by adding one
term that is an additional object, a new spatial relationship, or new attribute.
Do not add subjective descriptors to the prompt, it should be as factual as possible.
Do not use fluffy, poetic language, or any words beyond the elementary school level.
Every time you will be givng an existing prompt, write "Prompt:" and write the new
prompt, do not state anything else.

Existing prompt:  "a serene mountain lake with a small wooden dock and a small wooden
cabin on the shore"
Prompt:  a serene mountain lake with a small wooden dock and a small wooden cabin on
the shore, with birds in the sky

Existing prompt:  "elderly man wearing a backpack and a hat walking with a cane and a
dog on a leash"
Prompt:  elderly man wearing a backpack and a hat walking with a cane and a big white
dog on a leash

Existing prompt:  "a green fish hiding in a coral reef next to a crab"
Prompt:  a green fish hiding in a coral reef next to a crab and a purple octopus

Figure 19: System prompts for Llava-v1.6-34b when generating adaptive prompts, for each of the score bins
listed.

**Score** $[0.8, 1.0]$**:**
You are responsible for rewriting prompts for a computer program that generates
images.
You will be given an existing prompt that you must add complexity to by adding a
few terms.  Each term is an additional object, a new spatial relationship, or new
attribute.
Do not add subjective descriptors to the prompt, it should be as factual as possible.
Do not use fluffy, poetic language, or any words beyond the elementary school level.
Every time you will be givng an existing prompt, write "Prompt:" and write the new
prompt, do not state anything else.

Existing prompt:  "a serene mountain lake with a small wooden dock and a cabin on
the shore"
Prompt:  a serene mountain lake with a small wooden dock and a large cabin on the
shore, with birds in the sky and a fisherman fishing on the dock

Existing prompt:  "elderly man wearing a backpack and a hat walking with a cane and a
dog on a leash"
Prompt:  elderly man wearing a blue backpack, a red hat, and sunglasses walking with
a cane and a big white dog on a leash

Existing prompt:  "a green fish hiding in a coral reef next to a crab"
Prompt:  a striped green fish hiding in a coral reef next to a crab and a purple
octopus with the open ocean in the background

Existing prompt:  "a lemon tree on a hill"
Prompt:  a big lemon tree on a hill with a dog sitting underneath

Figure 20: System prompts for Llava-v1.6-34b when generating adaptive prompts, for the score bin listed.

## G   Ablations on Prompt Iterations

MT2IE generates a configurable number of prompts: in the Iterative Difficulty setting discussed in Section 5 prompts are progressively more difficulty in each iteration, whereas in the Adaptive Prompting setting discussed in Section 6 each iteration's prompt is rewritten by the MLLM based on the previous prompt's alignment score.

Rankings of 8 T2I models for prompt iterations={2, 5, 10, 12} for Iterative Difficulty are shown in Figure 21 and Adaptive Prompting in Figure 22. Additionally we calculate Kendall's $\tau$ rank correlation coefficient for all T2I model rankings, using the ranking obtained from GenAIBench as the ground truth ranking, shown in Table 6. We found that using 5 prompt iterations, *i.e.*, 20 total prompts, with MT2IEyields strongly-correlated model scores with the GenAIBench benchmark which contains 1600 prompts. 5 prompt iterations results in the highest ranking correlation and smaller standard deviations in model rankings. We suspect that using more prompt iterations in the Iterative Difficulty setting can result in prompts that are too hard for many T2I models, resulting in uninformative scores and model rankings.

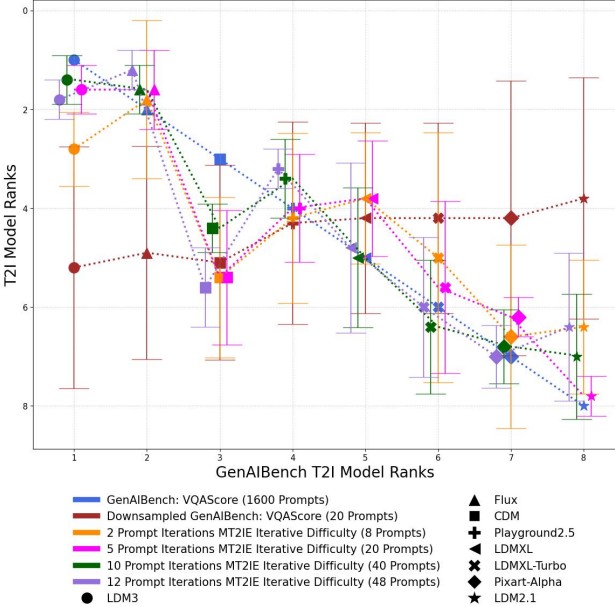

Figure 21: Average rank of 8 T2I models (shown on x axis where 1=highest scoring, 8=lowest scoring) evaluated by MT2IEwith progressively difficult prompt generation with various prompt iterations. All prompt iteration counts yield more informative rankings that better match GenAIBench than downsampled GenAIBench.

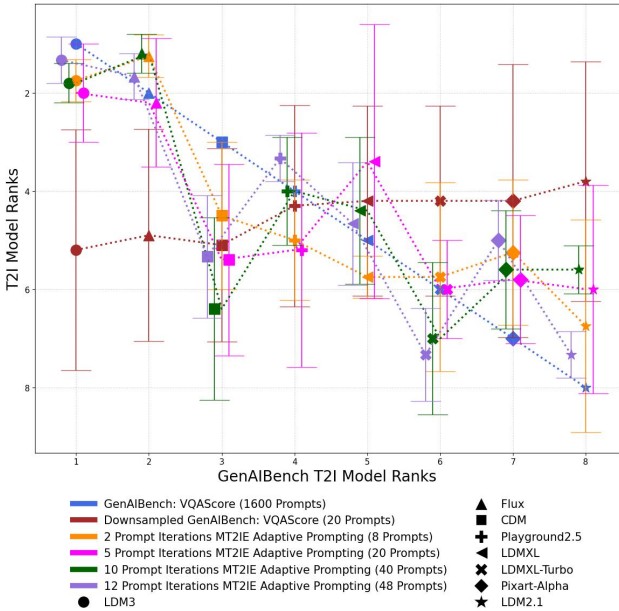

Figure 22: Average rank of 8 T2I models (shown on x axis where 1=highest scoring, 8=lowest scoring) evaluated by MT2IEwith adaptive prompt generation with various prompt iterations. All prompt iteration counts yield more informative rankings that better match GenAIBench than downsampled GenAIBench.

Table 6: Kendall's $\tau$ rank correlations between T2I model rankings produced by MT2IEfor all prompt iteration values and the full, 1600 prompt GenAIBench benchmark. We show the average $\tau$ between MT2IEwith Iterative Difficulty and Adaptive Prompting for each prompt iteration value, and compare to the ranking produced by downsampling GenAIBench to the same number of prompts as our highest-correlated prompt iteration value. MT2IEwith all prompt iteration values produces rankings notably more correlated to GenAIBench than downsampled GenAIBench.

| T2I Evaluation Method | Kendall's $\tau$ |
|---|---|
| Downsampled GenAIBench (20 Prompts) | $-0.8693$ |
| MT2IE: 2 Prompt Iterations (8 Prompts) | $0.6672$ |
| MT2IE: 5 Prompt Iterations (20 Prompts) | $0.7275$ |
| MT2IE: 10 Prompt Iterations (40 Prompts) | $0.7004$ |
| MT2IE: 12 Prompt Iterations (48 Prompts) | $0.7025$ |

## H  Comparing TIFAv1.0 and the Llava-Generated Question Set from TIFAv1.0 Prompts

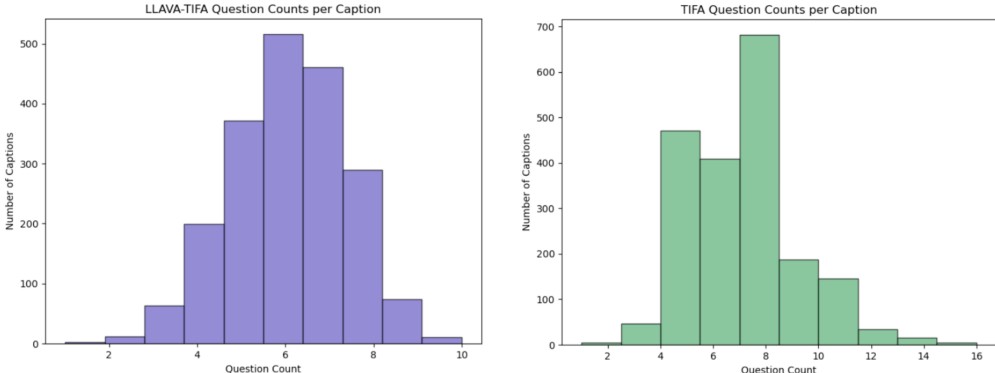

Figure 23: Element mismatch counts for all TIFAv1.0 4,081 captions (prompts). Most captions have just 1 or no element mismatch, indicating that Llava can break down and generate questions for prompts with similar performance to the proprietary GPT-3.

We analyze and compare TIFAv1.0 and our Llava-Generated question set (Llava-Gen TIFA) for the same 4,081 prompts from TIFA. This is done to verify that Llava is capable of generating questions as effectively as proprietary LLMs (TIFAv1.0 uses GPT-3 to generate questions).

**General question set statistics**  Overall, TIFAv1.0 has an average of 6.87 questions per prompt and Llava-Gen TIFA has an average of 6.14 questions per prompt. This results in Llava-Gen TIFA containing around 10% less questions than TIFAv1.0 while covering the same number of prompts. Despite this, Llava-Gen TIFA achieves higher correlations with human judgment than TIFAv1.0, supporting our claims that MLLM-generated T2I benchmarks are more efficient. The distribution of question counts per prompt are shown in Figure 23.

**Similarity of question elements**  TIFAv1.0 provides entity tags for each of their prompts, indicating which prompt elements the generated questions cover (Hu et al., 2023). We prompt Llava to similarly identify elements when generating questions and analyze the elements captured in TIFAv1.0 vs. Llava-Gen TIFA.

Across all prompts, TIFAv1.0 has an average of 4.68 elements per prompt and Llava-Gen TIFA has 4.13 elements per prompt, indicating an extremely similar number of elements identified for each prompt. For prompts with mismatched elements, the average number of mismatched elements is only 1.13, and we show counts of prompts with mismatched elements in Figure 24. The average BLEU (Papineni et al., 2002) score between TIFAv1.0 and Llava-Gen TIFA questions about the same elements is 0.65, which indicates a high level of similarity between questions covering the same concepts.

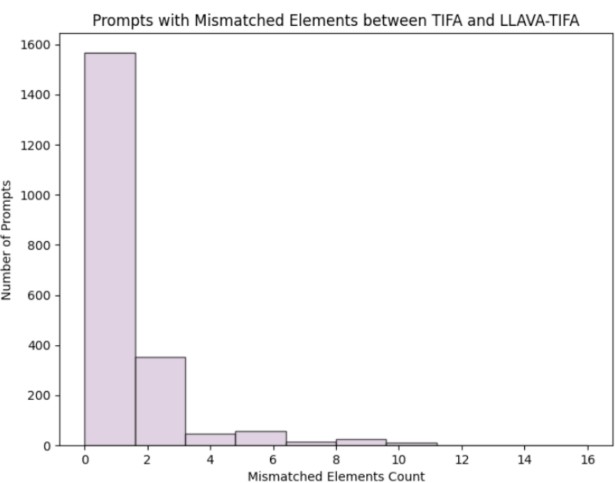

Figure 24: Element mismatch counts for all TIFAv1.0 4,081 captions (prompts). Most captions have just 1 or no element mismatch, indicating that Llava can break down and generate questions for prompts with similar performance to the proprietary GPT-3.

# I  Limitations & Future Work Discussion

The use of open-source MLLMs as dynamic evaluators for T2I models introduces several limitations to the evaluation process, inherited from various well-known weaknesses of MLLMs. Notably, MLLMs are susceptible to hallucination and implicit bias, generating outputs that appear credible but are not faithfully grounded in the input images Huang et al. (2024); Li et al. (2023). This is largely due to their tendency to over-rely on language patterns and statistical correlations inherent in training data, rather than genuine reasoning and understanding of their inputs. This has been systematically observed to undermine MLLM reliability and performance in tasks such as visual question answering and vision-language reasoning, which are crucial to using MLLMs as evaluators Goyal et al. (2017b); Thrush et al. (2022b); Tong et al. (2024). Recent work to mitigate MLLM hallucinations involves fine-tuning models on hallucination-corrected multimodal datasets or contrastive learning to discourage hallucinated text Huang et al. (2024); Jiang et al. (2024a). Both approaches have yielded significant improvements in hallucination rates when applied to finetuning Llava, the MLLM used in MT2IE. Because MT2IE is designed to work with any MLLM, improvements in reducing bias and hallucinations in these models can be directly leveraged. As more robust and reliable MLLMs become available, MT2IE can seamlessly incorporate them to deliver more trustworthy T2I evaluations.

Future work could explore leveraging multiple MLLMs as ensembles of T2I evaluators, to investigate whether majority voting among different models leads to more reliable evaluations; inspired by the use of inter-annotator agreement as a consistency measure for human evaluations of T2I models. Furthermore, as advanced reasoning techniques like chain-of-thought become available in multimodal models, incorporating self-reflection and self-verification within the T2I evaluation process could help validate image-text judgments and facilitate the creation of more effective, adaptive evaluation prompts Deng et al. (2025); Li et al. (2025).

