# OpenReview forum: "Multi-Modal Language Models as Text-to-Image Model Evaluators"
_TMLR — Rejected by TMLR_

### Review · Reviewer_SVTP · 2025-07-05

**Summary Of Contributions:**

This paper proposes a new evaluation framework for text-to-image (T2I) models called MT2IE. The core idea is to use an open-source multimodal large language model (MLLM) to generate prompts for iterative evaluation autonomously. For example, MLLM can gradually increase the difficulty of text prompts to assess how well a T2I model maintains text-image consistency. MT2IE aims to effectively replace traditional static benchmarks or VQA-based evaluation procedures. The authors empirically show that MT2IE can recover the relative order (ranking) of various T2I models using only 20 generated prompts, comparable to the previous approach that requires 1600 prompts.

**Audience:**

Yes

**Broader Impact Concerns:**

The paper does not explicitly discuss limitations or ethical concerns. However, some important points should be addressed. Using an MLLM to dynamically evaluate T2I models exposes the evaluation process to the implicit biases and errors of the MLLM itself. It would be essential to discuss these potential risks and possible future work.

**Claims And Evidence:**

No

**Requested Changes:**

Please see the weaknesses above.
* In Section 5.1 on aesthetic evaluation, adding an illustrative example (as shown in Figures 2 and 5) would improve clarity.
* Figure 7 is difficult to interpret. Consider splitting it into subfigures or redesigning it for better readability.

**Strengths And Weaknesses:**

**Strengths**

* The motivation is timely and relevant. As T2I models are rapidly evolving, concerns and potential weaknesses about the fixed and static benchmarks are valid.
* From the perspective of capturing relative model ranking, the reduction in the number of prompts needed provides a significant advantage in evaluation efficiency.
* The experimental structure (Experiment 1 -> 2 -> 3) is clear and logically progressive.

**Weaknesses**

* The work begins with a critique of static benchmarks but then emphasizes that MT2IE can achieve a similar relative ranking to such benchmarks. This is somewhat confusing… Moreover, since it leverages existing (and external) scoring methods like VQAScore, does this suggest MT2IE lacks an independent mechanism for assessing text-image correlation?
* The paper does not discuss much about the limitations of open-source MLLMs, such as hallucination, implicit bias, and data leakage. These issues can compromise the consistency and reliability of evaluations, especially given that MLLMs quickly become outdated these days.
* Although the authors try to justify the choice of Llava-1.6-34B, it remains unclear whether other MLLMs fail to get the ‘correct’ relative rankings. Does the consistency of results hold across different MLLMs? Empirical validation on this point would greatly strengthen the paper.
* While the paper claims that only 20 prompts are ‘sufficient’, it is important to show whether increasing the number of prompts (i.e., scaling) improves ranking stability or score difference. For example, as Figure 2 suggests, model scores would not increase consistently with more iterations.
* The experimental setup includes only 4 topic categories and 4 seed prompts. Given the huge diversity of images that T2I models can generate, this seems highly limited.
* Although the paper shows that MT2IE produces similar rankings to GenAIBench, it does not compare against the rankings captured by other previous approaches.
* As can be observed in Figure 4, the relative order is preserved, but the distances between each score differ significantly. This implies that MT2IE may capture model order but fail to reflect meaningful differences. In other words, the score gap also matters.

---

> ### Author Response · Authors · 2025-08-05
> **Response to Reviewer SVTP**
>
> **Clarifications Regarding Weaknesses**
> > The work begins with a critique of static benchmarks...
>
> We acknowledge that using static benchmarks as a baseline may seem confusing given our critiques of them. While we critique static benchmarks for their rigidity, we still use them as a baseline because there are no dynamic benchmarks available for direct comparison. Our key contribution is demonstrating that dynamically generating prompts can yield similar model rankings using significantly fewer prompts, supporting our argument that many prompts in static benchmarks are redundant or ineffective for meaningful evaluation.
>
> Good question about the use of existing scoring methods like VQAScore. Our framework contributes a novel way to both generate and score prompts dynamically; while we use VQAScore, our approach is designed to be flexible and support any scoring metric. We view this as a key strength – as scoring methods are introduced and improved, they can be used directly in MT2IE. Any scoring approach leveraging a MLLM can be used with MT2IE, and the results in Section 4 validate that a single MLLM is capable of matching or even surpassing human correlation in evaluating prompt-image consistency using established scoring methods like VQAScore.
>
> > While the paper claims that only 20 prompts are ‘sufficient’..
>
> We appreciate your insightful concern regarding prompt quantity and progressions. We would like to note that in Appendix G, we show that increasing the number of total prompts beyond 20—by generating more prompt iterations per seed prompt—actually leads to decreased ranking stability and higher standard deviation in model scores across different seeds. We hypothesize that scaling the number of prompts beyond 20 produces prompts that are excessively challenging for many T2I models, resulting in less informative scores and less reliable rankings.
>
> > The experimental setup includes only 4 topic categories and 4 seed prompts...
>
> For starting or seed prompts, we chose to generate 4 prompts spanning 4 categories that encompass COCO’s captions: household scenes (foods, household items, or furniture), descriptions of people, scenes with animals, and location descriptions. Empirically we found that allowing the MLLM to generate prompts of these categories over multiple seeds yields better T2I model average scores than starting from a subsample of COCO prompts. We have added these clarifications to the main paper.
> We acknowledge that our prompt diversity is lower than that of larger prompt sets. However, our approach aims to cover the various categories in COCO to ensure we capture at least the same range of topics. A promising direction for future work would be to find ways to maximize prompt diversity while keeping the number of prompts low for more efficient evaluation.
>
> > Although the paper shows that MT2IE produces similar rankings to GenAIBench, it does not compare against the rankings captured by other previous approaches.
>
> We appreciate your point regarding additional comparisons. We are currently running experiments to compare MT2IE’s rankings to rankings from the widely used GenEval benchmark [1], which will be finished by the camera-ready.
>
> [1] Dhruba Ghosh, et. al. GENEVAL: An object-focused framework for evaluating text-to-image alignment. arXiv preprint arXiv:2310.11513, 2023
>
> > As can be observed in Figure 4, the relative order is preserved, but the distances between each score differ significantly...
>
> We acknowledge that score gaps between T2I models are important for evaluation. Using rank correlation metrics to evaluate the relative ordering is a widely accepted method for validating evaluation approaches and benchmarks, so we focus on relative order to evaluate our work. Formal quantitative measures that jointly capture both correct ranking and meaningful score gaps remain underexplored, posing a challenge for score gap evaluation. As shown in Figure 4, MT2IE attains the closest ranking alignment compared to other methods, underscoring its strength in order preservation. Developing approaches to effectively incorporate score gaps alongside rankings is an important and promising direction for future work.
>
> **Actions Regarding Weaknesses**
>
> We have run new experiments using MT2IE with 2 additional MLLMs, with results in Appendix E. MT2IE’s rank correlations are consistently higher than those of existing evaluation methods across all MLLMs used, though rank correlation improves as MLLM size increases. If you would like to see additional results with a particular MLLM or model family, please let us know and we can provide the results in the camera-ready version.
>
> Section 5.1 has been moved to Appendix A, as aesthetic evaluation is not a main claim of our paper. We will add an illustrative example to this Appendix section.
>
> We are considering how to reformat Figure 7 so it’s more parsable.
>
> We have added a discussion on the limitations of MLLMs and how they may affect MT2IE, in Appendix I.

---

> > ### Author Response · Authors · 2025-08-07
> > **Figure Reformatting Completed**
> >
> > We have reorganized Figure 7 and its corresponding Appendix C Figure 9 into subfigures to improve clarity and readability.
> >
> > The latest PDF revision includes these new figures.

---

> > > ### Comment · Reviewer_SVTP · 2025-08-13
> > >
> > > Thank you for the additional experiments and further revisions. I will reflect the responses and the updated manuscript in my final recommendation.

---

### Review · Reviewer_Rmgw · 2025-07-08

**Summary Of Contributions:**

This paper introduces MT2IE, a unified framework for evaluating text-to-image (T2I) models using a single open-source MLLM to both generate prompts and assess image-prompt consistency. It supports iterative prompting and adaptive prompting, with a difficulty-weighted scoring method for fair comparison. The framework achieves model rankings comparable to large-scale human benchmarks using only ~20 automatically generated prompts.

**Audience:**

Yes

**Claims And Evidence:**

Yes

**Requested Changes:**

1. **Expand analysis of prompt set design**
The paper relies on a fixed set of four manually selected seed prompts, each expanded into a total of 20 prompts. This design may introduce topical bias and limit semantic coverage. While the paper ablates the number of iterations, it does not test how evaluation performance varies with more or more diverse seed prompts, nor whether results saturate beyond 20 prompts. Please consider analyzing the impact of prompt set size and diversity—e.g., by sampling or clustering from GenAIBench—to better support the robustness and generalizability of the chosen evaluation setup.
2. **(optional) Discuss or prototype error-aware adaptation**
The current adaptive prompting strategy adjusts difficulty based on global consistency scores but does not respond to specific types of generation failures (e.g., missing objects or spatial mistakes). Including a discussion—or an exploratory example—of how targeted, error-aware prompt adjustments could enhance evaluation fidelity would strengthen the alignment with real user behavior.

**Strengths And Weaknesses:**

**Strengths**

1. **Well-structured and progressively motivated experimental design**
    The paper presents three stages of evaluation: reproducing static benchmark rankings, probing model capabilities with iterative prompting, and introducing adaptive prompting with difficulty-aware scoring. This progression reflects strong experimental design and builds trust in the framework.

2. **An effective and efficient unified evaluation framework**
    MT2IE uses a single MLLM to perform prompt generation, scoring, and evaluation. It matches benchmark performance using only ~20 prompts, all generated automatically—eliminating the need for handcrafted datasets and enabling scalable, low-cost evaluation.

3. **Careful handling of adaptive evaluation bias**
    The framework addresses the challenge that stronger models may receive harder prompts by weighting consistency scores with prompt difficulty. This ensures fair model comparison even under personalized prompting.

**Limitations and Future Directions**

1. **Potential bias from limited seed prompt diversity**
    The framework uses a fixed set of four manually selected seed prompts, each expanded through a predefined number of iterations to produce 20 evaluation prompts. While this setup is efficient, relying on a small and fixed seed set may introduce topical or structural bias, limiting the semantic coverage of the evaluation. While the paper ablates the number of iterations per seed prompt, it keeps the seed set fixed across all experiments. As a result, it does not examine how evaluation performance might vary with a larger or more diverse seed set. For example, clustering or summarizing prompts from large-scale datasets like GenAIBench (1600 prompts) could help construct a more representative seed set, while also better utilizing the human effort embedded in those benchmarks. A more systematic exploration of prompt initialization and scaling would strengthen the generalizability and robustness of the evaluation design.

2. **(Potential improvement) Lack of error-aware adaptation**
    The adaptive prompting mechanism adjusts difficulty based on overall scores but does not consider which specific elements of a prompt were poorly realized in the image. While the method works well as designed, introducing more targeted, error-aware adaptation—such as emphasizing missing objects or incorrect attributes—could enhance diagnostic value and bring the framework closer to human evaluation behavior.

---

> ### Author Response · Authors · 2025-08-05
> **Response to Reviewer Rmgw**
>
> **Clarifications and Actions**
> > 1. Expand analysis of prompt set design...
>
> We appreciate your insightful comment on seed prompt diversity and quantity. MT2IE generates 4 seed prompts spanning 4 categories that encompass COCO’s captions: household scenes (foods, household items, or furniture), descriptions of people, scenes with animals, and location descriptions. The prompt categories are manually selected based on COCO’s categories, but the starting prompts themselves differ based on the seed and MLLM’s sampling hyperparameters. Empirically we found that allowing the MLLM to generate prompts of these categories over multiple seeds yields better T2I model average scores than starting from a subsample of COCO prompts. We have added these clarifications to the main paper.
>
> Evaluating how the number of seed prompts influences performance, and whether seed prompts can be generated by clustering existing benchmarking prompt sets, is a promising area for future research. Additionally, it would be valuable to explore methods for maximizing text diversity among the initial prompts.
>
> > 2. Lack of error-aware adaptation...
>
> Thank you for your thoughtful suggestion highlighting the value of error-aware adaptation within the adaptive prompting mechanism. We completely agree that moving beyond global score-based adjustments to incorporate feedback on specific prompt-image misalignments could significantly enhance the diagnostic capability and interpretability of MT2IE.
> Extending MT2IE to automatically analyze and report on particular types of generation errors, and to adapt prompts in a more targeted manner based on those error patterns, is a promising future direction. We see this as both an exciting and natural extension of the current framework, with strong potential to bring the evaluation process closer to the nuanced, element-specific analysis typical of human reviewers.

---

### Review · Reviewer_qu7t · 2025-07-10

**Summary Of Contributions:**

The paper proposes a new framework for the automated evaluation of Text-to-Image (T2I) models, named Multimodal Text-to-Image Eval (MT2IE). The main contributions are as follows:
- MT2IE utilizes open-source multimodal large language models (MLLMs) without fine-tuning for the evaluation process. MLLMs serve as both prompt generators for T2I models and evaluators of their outputs. This eliminates the need for using multiple models for T2I evaluation and demonstrates that untuned, open-source MLLMs can provide a sufficiently strong signal that correlates well with human judgments.
- MT2IE employs an iterative procedure for generating prompts of increasing complexity to test T2I models on prompt consistency. This method matches the evaluation quality of large-scale static benchmark (GenAIBench) while using much fewer prompts, resulting in a significant improvement in evaluation efficiency.
- The authors propose an adaptive prompt generation method for evaluating prompt consistency. In this procedure, prompts are generated iteratively, with the complexity of each new prompt adjusted based on the model’s performance on previous ones. This enables the framework to identify specific scenarios in which a model fails to produce prompt-consistent images.
- The authors also demonstrate that MT2IE can be extended to evaluate image aesthetics, with its rankings closely aligned with those of widely-used LAION-based aesthetic scoring models.

**Audience:**

Yes

**Broader Impact Concerns:**

-

**Claims And Evidence:**

No

**Requested Changes:**

Based on the weaknesses outlined above, I would request the following:
- Do experiments analyzing the dependence of evaluation score correlations and their variances on the number of prompts used. Additionally, provide recommendations for the number of prompts (and other evaluation parameters, e.g., starting prompts), along with MT2IE results using these parameters for several T2I models. I believe this is crucial, as the paper proposes a new evaluation framework.
- Use at least one additional benchmark besides GenAIBench to test MT2IE.
- Provide results for MT2IE using smaller MLLMs.
- If the paper claims that MT2IE can be used for evaluating aspects beyond prompt consistency, this should be more thoroughly supported. For example, aesthetics scoring using an MLLM on images generated from prompts other than those produced by MT2IE could be tested. Additionally, an analysis of which tasks can be reliably evaluated using MLLMs should be included to support the claim that “MT2IE can be easily repurposed to evaluate axes other than prompt consistency by simply rewriting the MLLM prompt used for judgment.”

These would be nice to have, though not critical:
- Human evaluation on the benchmark generated by MT2IE to support the claim that MT2IE provides scores with high correlation to human judgment on its own dataset.
- Analysis of MLLM performance on complex prompts with varied syntactic structures.

**Strengths And Weaknesses:**

Strengths:
- The paper addresses the important problem of efficient evaluation of Text-to-Image (T2I) models. Most existing evaluation benchmarks require running inference on a large set of prompts and using one or more downstream models to assess the generated images, which is computationally expensive. Improving the efficiency of T2I model evaluation is therefore highly valuable. MT2IE produces evaluation metrics that correlate strongly with those from the current benchmark, GenAIBench, while using approximately 80 times fewer prompts.
- MT2IE is conceptually simple: it uses a single untuned model for end-to-end evaluation of T2I models.
- MT2IE leverages open-source models, making the framework accessible, transparent, and reproducible.
- MT2IE generates prompts of increasing complexity, enabling model rankings based on performance across prompts of varying difficulty.
- The authors propose an adaptive prompt generation method to evaluate T2I models on prompt consistency. Prompts are generated iteratively, with their complexity adjusted based on the model’s performance on previous prompts. This allows the identification of specific scenarios where each model fails to generate prompt-consistent images.
- MT2IE can also be used to evaluate image aesthetics, with its rankings closely aligned with those of widely-used LAION-based aesthetic scoring models.
- The paper provides strong quantitative analysis, showing that MT2IE scores align well with human judgments and current benchmark metrics.

Weaknesses and Nice-to-Haves
- Only one benchmark (GenAIBench) is used to compare MT2IE scores. here are other widely used ones, for example, COCO-Attribute[1], Animal-Scene[1]. It would be beneficial to assess how well MT2IE scores correlate with those on additional datasets. The observed correlation with GenAIBench may simply result from structural similarities between MT2IE-generated prompts and those in GenAIBench, especially given that the latter’s prompts were written by professional designers who may follow specific stylistic patterns.
- The prompts generated by MT2IE follow a specific syntactic structure. While this is useful for evaluating prompt consistency by including object relations, real-world prompts often describe similar scenes in more sophisticated or varied ways. This limits the applicability of MT2IE in real-life scenarios, may introduce bias in the evaluation process and may restrict its ability to fully uncover model weaknesses in the adaptive prompt setting.
- It would be valuable to better understand why MT2IE performs well. Specifically, why its MLLM-based evaluations correlate so highly with human judgments and GenAIBench scores. Does the MLLM attend to object count, spatial relationships, or only surface-level matching? Testing the MLLM on prompts with varied structures and examining attention maps could help reveal potential weaknesses or biases.
- Although the authors claim that MT2IE achieves higher correlation with human judgments than existing metrics on static benchmarks like TIFA and DSG, there is no human evaluation of the MT2IE-generated benchmark itself. Including such human annotations would support the claim that MT2IE scores align well with human preferences on its own dataset.
- In the adaptive prompt generation scenario, it’s unclear whether simplifying prompts based on poor model performance is well justified. If the goal is to expose model weaknesses, one might prefer to identify prompts where the model performs poorly rather than making tasks easier. Moreover, since prompts differ across models, the adaptive setting may not be suitable for standardized benchmarking, even if prompt complexity is weighted. That said, I do believe that adaptive generation of prompts can be highly beneficial for revealing weaknesses of models, if used in a proper scenario.
- MT2IE’s ability to evaluate image aesthetics is compared to only one scoring model. If aesthetics evaluation is a key contribution, more extensive experiments are needed. For instance, testing MT2IE on images generated by other prompts (not produced by MT2IE) could help determine whether the MLLM’s aesthetic judgments align with other methods and with human preferences.
- The paper claims that MT2IE can be repurposed to evaluate other axes (e.g., aesthetics, factuality) by simply changing the MLLM’s system prompt. However, this relies heavily on the capabilities of the underlying MLLM. A deeper understanding of the MLLM’s limitations in interpreting text and images would clarify which tasks MT2IE can realistically support.
- Figure 7 suggests that MT2IE evaluation results exhibit high variance (if I interpreted it correctly). In practice, this implies the need to run multiple evaluation trials or use more prompts to achieve stable results. The paper would benefit from an empirical study showing how evaluation score variance and correlation with human judgments change with the number of prompts. As MT2IE proposes a new benchmark, it should include recommended prompt counts and starting prompts along with evaluation results for several T2I models.
- All MT2IE analysis is based on a single MLLM (Llava-v1.6-34b), which is large and computationally expensive. It would be useful to report results using smaller MLLMs as well. Even if they are less accurate, they might still provide acceptable evaluation quality and offer a more resource-efficient option for lightweight T2I model testing.

[1] Yumeng Li, Margret Keuper, Dan Zhang, and Anna Khoreva. Divide & bind your attention for improved generative semantic nursing. arXiv preprint arXiv:2307.10864, 2023.

---

> ### Author Response · Authors · 2025-08-05
> **Response to Reviewer qu7t**
>
> **Clarifications and Actions**
> > If the paper claims that MT2IE can be used for evaluating aspects beyond prompt consistency, this should be more thoroughly supported….
>
> Thank you for your thoughtful feedback. We agree that more extensive evidence is necessary to substantiate the use of MT2IE for evaluating T2I aspects beyond prompt consistency. In response, we have moved Section 5.1 on aesthetic evaluation to Appendix A and have removed statements from the main text that present MT2IE as a general-purpose evaluation method for tasks beyond prompt consistency. The exploratory results on aesthetics evaluation were intended only to suggest possible broader applications, not to make definitive claims. We appreciate your recommendation regarding a systematic analysis of which tasks can be reliably evaluated with MLLMs, and we recognize this as an important direction for future work.
>
> > Provide results for MT2IE using smaller MLLMs.
>
> Good suggestion! We have run additional experiments using MT2IE with 2 smaller MLLMs, with results in Appendix E. Unsurprisingly, larger MLLMs yield better results but even with a 7 billion MLLM, MT2IE’s rank correlations are higher than existing evaluation methods. If you would like to see additional results with a particular MLLM or model family, please let us know and we can provide the results in the camera-ready version.
>
> > Do experiments analyzing the dependence of evaluation score correlations and their variances on the number of prompts used...
>
> We appreciate your emphasis on analyzing scores with respect to prompt count, as well as the need for concrete recommendations and results for these evaluation parameters.
> We would like to note that in Appendix G, we provide an ablation study on the prompt iteration hyperparameter in MT2IE with the experimental setups in Sections 5 and 6, reporting rankings of 8 T2I models for each hyperparameter value. Our results show that increasing the number of prompt iterations beyond 5 (i.e., using more than 20 total prompts with 4 starting prompts) leads to less stable rankings and higher standard deviation in model scores across different seeds. We hypothesize that increasing the number of prompt iterations beyond 5 generates overly challenging cases for many T2I models, resulting in less informative scores and less reliable rankings. Based on these findings, we set the number of prompts to 20 in our main paper experiments and recommend using 5 prompt iterations with MT2IE.
>
> For starting or seed prompts, we chose to generate 4 prompts spanning 4 categories that encompass COCO’s captions: household scenes (foods, household items, or furniture), descriptions of people, scenes with animals, and location descriptions. Empirically we found that allowing the MLLM to generate prompts of these categories over multiple seeds yields better T2I model average scores than starting from a subsample of COCO prompts. We have added these clarifications to the main paper.
>
> >Only one benchmark (GenAIBench) is used to compare MT2IE scores.
>
> We appreciate your thoughtful suggestion regarding the use of additional benchmarks beyond GenAIBench and agree that demonstrating MT2IE's effectiveness across multiple datasets would strengthen the validity of our correlation results. Unfortunately due to time constraints we could not run experiments on additional benchmarks by the end of this discussion period but will have these experiments in the camera-ready version.
>
> Regarding the recommended benchmarks, COCO-Attribute and Animal-Scene are constructed by complexifying or filtering existing prompt sets in [1]. To the best of our knowledge, neither the prompt sets themselves nor the construction code have been publicly released with the cited work, limiting our ability to evaluate MT2IE on them. If you are aware of a way to access them, we would be eager to include experiments on them in an updated version of our work.
>
> We are currently conducting experiments on an additional benchmark, GenEval [2], a widely used compositional text-to-image (T2I) benchmark with prompts based on COCO. These experiments will allow us to perform further rank correlation comparisons with MT2IE. Due to time constraints, we are unable to include these results in this response, but they will be ready for the camera-ready version.
>
> *[1] Yumeng Li, Margret Keuper, Dan Zhang, and Anna Khoreva. Divide & bind your attention for improved generative semantic nursing. arXiv preprint arXiv:2307.10864, 2023.*
> *[2] Dhruba Ghosh, Hannaneh Hajishirzi, and Ludwig Schmidt. GENEVAL: An object-focused framework for evaluating text-to-image alignment. arXiv preprint arXiv:2310.11513, 2023*

---

> > ### Comment · Reviewer_qu7t · 2025-08-22
> >
> > Thank you for your responses and edits, it is appreciated.
> >
> > "Our results show that increasing the number of prompt iterations beyond 5 (i.e., using more than 20 total prompts with 4 starting prompts) leads to less stable rankings and higher standard deviation in model scores across different seeds. We hypothesize that increasing the number of prompt iterations beyond 5 generates overly challenging cases for many T2I models, resulting in less informative scores and less reliable rankings" — it would be really interesting to investigate why exactly this leads to less stable rankings and higher standard deviation in model scores. This might be helpful in understanding failure modes of T2I models

---

### Author Response · Authors · 2025-07-16
**Request for Extension of Reviewer Response Period**

We thank the reviewers for their thoughtful input on our work.

We would like to request an extension of 2 weeks to the reviewer response period to adequately fulfill changes requested by reviewers. The current two week period coincides with ICML conference, limiting our time to respond.

---

### Author Response · Authors · 2025-08-05
**Overall Reviewer Response**

We sincerely thank all reviewers for their thoughtful evaluations and engagement with our work. We appreciate the recognition of MT2IE’s contributions in enabling a more efficient and adaptive evaluation of text-to-image (T2I) models, particularly noting our framework’s strong empirical alignment with current benchmarks and human judgments, its flexibility, and its conceptual simplicity. We are grateful for the acknowledgement of the rigorous experimental design and the potential of MT2IE to streamline T2I models’ prompt consistency evaluation.

Reviewer qu7t raised several points regarding the need for broader benchmark validation via experiment expansions including using smaller MLLMs, more thorough evaluation of aesthetic scoring, and potential inclusion of human annotation on MT2IE-generated datasets.
Reviewer Rmgw suggested a systematic exploration of MT2IE’s seed prompts to examine robustness and the possible benefits of error-aware adaptive prompting.
Reviewer SVTP highlighted concerns about the reliance on static benchmarks for validation and limitations relating to quantities of prompt and MLLMs used. Additionally SVTP brought up the necessity of clarifying MT2IE’s handling of ranking stability and meaningful score differences, as well as the risks associated with implicit bias in MLLMs and figure clarification.
We value these constructive insights and suggestions, which will help us clarify, extend, and further validate MT2IE.

We address each reviewer’s individual concerns in corresponding comment responses and explain the overall steps we have taken to strengthen our submission:

**Actions Regarding Reviewer’s Concerns**
* While our primary focus in this work is on evaluating prompt consistency, we included initial explorations of aesthetics evaluation to highlight the potential versatility of MT2IE. To better align with the paper’s core contributions, we have relocated Section 5.1 on aesthetics to Appendix A and have removed statements from the main text that present MT2IE as a general-purpose evaluation method for tasks beyond prompt consistency. This adjustment clarifies our key contributions, while still providing interested readers with the aesthetics evaluation results for reference.

* We have performed additional experiments using MT2IE with more MLLMs, specifically spanning a wider range of model sizes, with results in Appendix E. Results show that rank correlation scales with MLLM size, with larger MLLMs yielding more aligned T2I model rankings and lower variance. However, MT2IE’s rank correlations are higher than existing evaluation methods even when used with the smallest, 7-billion MLLM, showcasing that our results hold across MLLM sizes.
If the reviewers would like to see additional results with a particular MLLM or model family, please let us know and we can provide the results in the camera-ready version.

* We are currently conducting experiments to compare MT2IE’s model rankings with an additional T2I benchmark: GenEval [1], a widely adopted compositional T2I benchmark. The results of these additional analyses will be ready for the camera-ready version.

* A discussion of the limitations of MLLMs, their impact on the MT2IE framework, and potential future directions to address these issues has been added to Appendix I.
* We have clarified in the main paper text how seed prompts are generated and specified that their topics were carefully chosen to represent the full range of COCO’s categories.

[1] Dhruba Ghosh, Hannaneh Hajishirzi, and Ludwig Schmidt. GENEVAL: An object-focused framework for evaluating text-to-image alignment. arXiv preprint arXiv:2310.11513, 2023

---

### Decision · Action_Editor_S4R5 · 2025-09-08

**Recommendation:** Reject

**Additional Comments:**

I think that the method might benefit from being fleshed out much more thoroughly. Some reviewers expressed concerns about the limited experiments and suggested comparisons beyond GenAIBench, additional experiments to validate the robustness of the method, and specific improvements to clarity of writing (all of which, I agree, would meaningfully improve the paper). Unfortunately, the authors did not address many of these points during the rebuttal phase --- despite the extra 2 weeks provided for the rebuttal.

Moreover, while TMLR allows for an "accept/minor revision" decision, the variance in the reported scores are high (according to my understanding of Figure 7) and the addition of other T2I benchmarks (e.g. GENEVAL) will be significant enough that any new interesting findings or discrepancies need to be carefully scrutinized. I feel that even if successfully executed this rises above the bar of a minor revision.

I look forward to receiving the revised version from the authors.

**Audience:**

Yes

**Audience Explanation:**

The paper proposes a method to benchmark text-to-image generative models that can be potentially of interest to several computer vision and machine learning researchers.

**Claims And Evidence:**

No

**Claims Explanation:**

The paper introduces MT2IE, a simplified evaluation framework for text-to-image (T2I) generative models. The main idea is to use a multimodal LLM to iteratively generate progressively more difficult prompts, and use the same LLM to score the generated images from the T2I model under consideration.

From reading the paper, the central hypothesis seems to be that a single multimodal LLM is as good as a more complicated workflow (involving a combination of prompt generator LLMs and VQA models) for benchmarking the capabilities of T2I generators. The authors provide a few experiments as evidence in support of this hypothesis.

The main issue as pointed out by some of the reviewers seems to be that the experiments need to be more convincing, and the writing needs to be clearer. Two reviewers raised the issue that the only results involved rank-correlations with a single existing benchmark (GenAIBench), and it would be helpful to ensure that the method generalizes more robustly across benchmarks. Reviewers also pointed out that increasing the prompt iterations (counter-intuitively) decreased correlation with human evals, raising questions about robustness of the presented experiments to other settings. The clarity of the presented results also needs to be improved (for example, Appendix G and Figures 21/22 continue to be confusing).

The authors mentioned in their latest response that they are still working on some of these aspects, but I feel that this may require a major revision; see the box below for further comments.

**Resubmission Of Major Revision:**

The authors may consider submitting a major revision at a later time.